# Rectifying the Shortcut Learning of Background for Few-Shot Learning

**Xu Luo[1], Longhui Wei[2], Liangjian Wen[1], Jinrong Yang[4], Lingxi Xie[3],**
**Zenglin Xu[6,7*], Qi Tian[5*]**
[1]University of Electronic Science and Technology of China
[2]University of Science and Technology of China [3]Tsinghua University
[4]Huazhong University of Science and Technology [5]Xidian University
[6]Harbin Institute of Technology Shenzhen [7]Pengcheng Laboratory
Frank.Luox@outlook.com,{weilh2568,zenglin}@gmail.com

## Abstract

The category gap between training and evaluation has been characterised as one of the main obstacles to the success of Few-Shot Learning (FSL). In this paper, we for the first time empirically identify image background, common in realistic images, as a shortcut knowledge helpful for in-class classification but ungeneralizable beyond training categories in FSL. A novel framework, COSOC, is designed to tackle this problem by extracting foreground objects in images at both training and evaluation without any extra supervision. Extensive experiments carried on inductive FSL tasks demonstrate the effectiveness of our approaches.

## 1 Introduction

Through observing a few samples at a glance, humans can accurately identify brand-new objects. This advantage comes from years of experiences accumulated by the human vision system. Inspired by such learning capabilities, Few-Shot Learning (FSL) is developed to tackle the problem of learning from limited data [24, 53]. At training, FSL models absorb knowledge from a large-scale dataset; later at evaluation, the learned knowledge is leveraged to solve a series of downstream classification tasks, each of which contains very few support (training) images from *brand-new categories*.

The category gap between training and evaluation has been considered as one of the core issues in FSL [10]. Intuitively, the prior knowledge of *old* categories learned at training may not be applicable to *novel* ones. [62] consider solving this problem from a causal perspective. Their backdoor adjustment method, however, adjusts the prior knowledge in a black-box manner and cannot tell which specific prior knowledge is harmful and should be suppressed.

In this paper, we identify image background as one specific harmful source knowledge for FSL. Empirical studies in [56] suggest that there exists spurious correlations between background and category of images (*e.g.*, birds usually stand on branches, and shells often lie on the beaches; see Fig. 1), which serves as a shortcut knowledge for modern CNN-based vision systems to learn. It is further revealed that background knowledge has positive impact on the performance of in-class classification tasks. As illustrated in the simple example of Fig. 1, images from the same category are more likely to share similar background, making it possible for background knowledge to generalize from training to testing in common classification tasks. For FSL, however, the category gap produces brand-new foreground, background and their combinations at evaluation. The correlations learned at training thus may not be able to generalize and would probably mislead the predictions. We take

---

*Corresponding author
Code: https://github.com/Frankluox/FewShotCodeBase

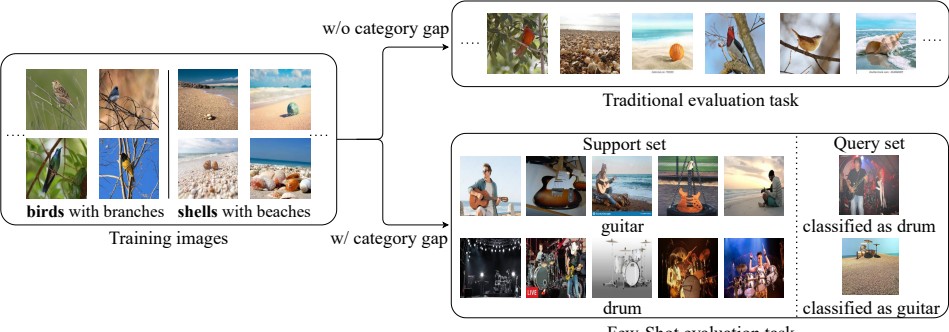

Figure 1: An illustrative example that demonstrates why background information is useful for regular classification but harmful for few-shot learning.

empirical investigations on the role of image foreground and background in FSL, revealing how image background drastically affects the learning and evaluation of FSL in a negative way.

Since the background is harmful, it would be good if we could force the model to concentrate on foreground objects at both training and evaluation, but this is not easy since we do not have any prior knowledge of the entity and position of the foreground objects in images. When humans are going to recognize foreground objects of images from the same class, they usually look for a shared local pattern that appears in the majority of images, and recognize patches with this pattern as foreground. This inspires us to design a novel framework, COSOC, to extract foreground of images for both training and evaluation of FSL by seeking shared patterns among images. The approach does not depend on any additional fine-grained supervisions such as bounding boxes or pixel-level labelings.

The procedure of foreground extraction of images in the training set is implemented before training. The corresponding algorithm, named **C**lustering-based **O**bject **S**eeker (COS), first pre-trains a feature extractor on the training set using contrative learning, which has an outstanding performance, shown empirically in a later section, on the task of discriminating between ground-truth foreground objects. The feature extractor then maps random crops of images—candidates of foreground objects—into a well-shaped feature space. This is followed by runing a clustering algorithm on all of the features of the same class, imitating the procedure of seeking shared local patterns inspired by human behavior. Each cropped patch is then assigned a foreground score according to its distance to the nearest cluster centroid, for determining a sampling probability of that patch in the later formal training of FSL models. For evaluation, we develop **S**hared **O**bject **C**oncentrator (SOC), an algorithm that applies iterative feature matching within the support set, looking for one crop per image at one time that is most likely to be foreground. The sorted averaging features of obtained crops are further leveraged to match crops of query images so that foreground crops have higher matching scores. A weighted sum of matching scores are finally calculated as classification logits of each query sample. Compared to other potential foreground extracting algorithms such as saliency-based methods, our COS and SOC algorithms have additional capability of capturing shared, inter-image information, performing better in complicated, multi-object scenery. Our methods also have flexibility of dynamically assigning beliefs (probabilities) to all candidate foreground objects, relieving the risk of overconfidence.

Our contributions can be summarized as follows. *i*) By conducting empirical studies on the role of image foreground and background in FSL, we reveal that image background serves as a source of shortcut knowledge which harms the evaluation performance. *ii*) To solve this problem, we propose COSOC, a framework combining COS and SOC, which can draw the model's attention to image foreground at both training and evaluation. *iii*) Extensive experiments for non-transductive FSL tasks demonstrate the effectiveness of our method.

## 2   Related Works

**Few-shot Image Classification.** Plenty of previous work tackled few-shot learning in meta-learning framework [18, 50], where a model learns experience about how to solve few-shot learning tasks by tackling pseudo few-shot classification tasks constructed from the training set. Existing methods that ultilize meta-learning can be generally divided into three groups: (1) Optimization-based methods learn the experience of how to optimize the model given few training samples. This kind of methods

either meta-learn a good model initialization point [12, 45, 39, 70, 20] or the whole optimization process [40, 58, 34, 27] or both [3, 36]. (2) Hallucination-based methods [16, 54, 46, 67, 25, 9, 26, 37] learn to augment similar support samples in few-shot tasks, thus can greatly alleviate the low-shot problem. (3) Metric-based methods [53, 48, 49, 61, 59] learn to map images into a metric feature space and classify query images by computing feature distances to support images. Among them, several recent works [19, 63, 57, 10] intended to seek correspondence between images either by attention or meta-filter, in order to obtain a more reasonable similarity measure. Our SOC algorithm in one-shot setting is in spirit similar to these methods, in that we both apply pair-wise feature alignment between support and query images, implicitly removing backgrounds that are more likely to be dissimilar across images. SOC differs in multi-shot setting, where potentially useful shared inter-image information in support set exists and can be captured by our SOC algorithm.

**The Influence of Background.** A body of prior work studied the impact of image background on learning-based vision systems from different perspectives. [52] showed initial evidence of the existence of background correlations and how it influences the predictions of vision models. [64, 43] analyzed background dependence for object detection. Another relevant work [4] utilized camera traps for investigating how performance drops when adapting classifiers to unseen cameras with novel backgrounds. They explore the effect of class-independent background (i.e., background changes from training to testing while categories remain the same) on classification performance. Although the problem is also concerned with image background, no shortcut learning of background exists under this setting. This is because under each training camera trap, the classifier must distinguish different categories with background fixed, causing the background knowledge being not useful for predictions of training images. Instead, the learning signal during training pushes the classifier towards ignoring each specific background. The difficulties under this setting lie in the domain shift challenge—the classifier is confident to handle previously existing backgrounds, but lost in novel backgrounds. More recently, [56] systematically explore the role of image background in modern deep-learning-based vision systems through well-designed experiments. The results give clear evidence on the existence of background correlations and identify it as a *positive* shortcut knowledge for models to learn. Our results, on the contrary, identify background correlations as a *negative* knowledge in the context of few-shot learning.

**Contrastive Learning.** Recent success on contrastive learning of visual representations has greatly promoted the development of unsupervised learning [6, 17, 15, 5]. The promising performance of contrastive learning relies on the instance-level discrimination loss which maximizes agreement between transformed views of the same image and minimizes agreement between transformed views of different images. Recently there have been some attempts [30, 13, 10, 33, 35, 31] at integrating contrastive learning into the framework of FSL. Although achieving good results, these work struggle to have an in-depth understanding of why contrastive learning has positive effects on FSL. Our work takes a step forward, revealing the advantages of contrastive learning over supervised FSL models in identifying core objects of images.

# 3 Empirical Investigation

**Problem Definition.** Few-shot learning consists of a training set $\mathcal{D}_B$ and an evaluation set $\mathcal{D}_v$ which share no overlapping classes. $\mathcal{D}_B$ contains a large amount of labeled data and is usually used at first to train a backbone network $f_\theta(\cdot)$. After training, a set of $N$-way $K$-shot classification tasks $\mathcal{T} = \{(\mathcal{S}_\tau, \mathcal{Q}_\tau)\}_{\tau=1}^{N_\mathcal{T}}$ are constructed, each by first sampling $N$ classes in $D_v$ and then sampling $K$ and $M$ images from each class to constitute $\mathcal{S}_\tau$ and $\mathcal{Q}_\tau$, respectively. In each task $\tau$, given the learned backbone $f_\theta(\cdot)$ and a small support set $\mathcal{S}_\tau = \{(x_{k,n}^\tau, y_{k,n}^\tau)\}_{k,n=1}^{K,N}$ consisting of $K$ images $x_{k,n}^\tau$ and corresponding labels $y_{k,n}^\tau$ from each of $N$ classes, a few-shot classification algorithm is designed to classify $MN$ images from the query set $\mathcal{Q}_\tau = \{(x_{mn}^\tau)\}_{m,n=1}^{M,N}$.

**Preparation.** To investigate the role of background and foreground in FSL, we need ground-truth image foreground for comparison. However, it is time-consuming to label the whole dataset. Thus we select only a subset $\mathcal{D}_{\text{new}} = (\mathcal{D}_B, \mathcal{D}_v)$ of *mini*ImageNet [53] and crop each image manually according to the largest rectangular bounding box that contains the foreground object. We denote the uncropped version of the subset as $(\mathcal{D}_B\text{-Ori}, \mathcal{D}_v\text{-Ori})$, and the cropped foreground version as $(\mathcal{D}_B\text{-FG}, \mathcal{D}_v\text{-FG})$. Two well-known FSL baselines are selected in our empirical studies: Cosine

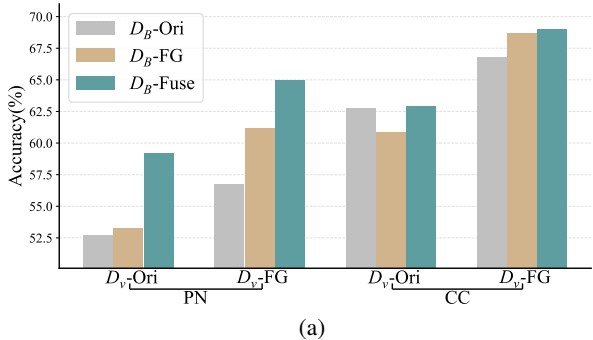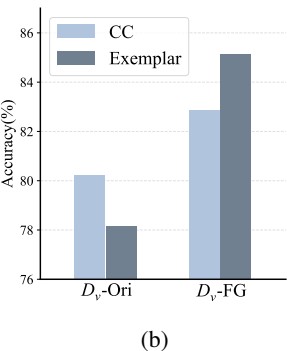

| (a) | (b) |

Figure 2: **5-way 5-shot FSL performance on different variants of training and evaluation datasets detailed in Sec. 3.** (a) Empirical exploration of image foreground and background in FSL using two models: PN and CC. (b) Comparison between CC and Exemplar trained on the full training set of *mini*ImageNet and evaluated on $\mathcal{D}_v$-Ori and $\mathcal{D}_v$-FG.

Classifier (CC) [14] and Prototypical Networks (PN) [48]. See Appendix A for details of constructing $\mathcal{D}_{new}$ and a formal introdcution of CC and PN.

## 3.1 The Role of Foreground and Background in Few-Shot Image Classification

Fig. 2(a) shows the average of 5-way 5-shot classification accuracy obtained by training CC and PN on $\mathcal{D}_B$-Ori and $\mathcal{D}_B$-FG, and evaluating on $\mathcal{D}_v$-Ori and $\mathcal{D}_v$-FG, respectively. See Appendix F for additional 5-way 1-shot experiments.

**Category gap disables generalization of background knowledge.** It can be first noticed that, under any condition, the performance is consistently and significantly improved if background is removed at the evaluation stage (switch from $\mathcal{D}_v$-Ori to $\mathcal{D}_v$-FG). The result implies that background at the evaluation stage in FSL is harmful. This is the opposite of that reported in [56] which shows background helps improve on the performance of traditional classification task, where no category gap exists between training and evaluation. Thus we can infer that the class/distribution gap in FSL disables generalization of background knowledge and degrades performance.

**Removing background at training prevents shortcut learning.** When only foreground is given at evaluation ($\mathcal{D}_v$-FG), the models trained with only foreground ($\mathcal{D}_B$-FG) perform much better than those trained with original images ($\mathcal{D}_B$-Ori). This indicates that models trained with original images may not pay enough attention to the foreground object that really matters for classification. Background information at training serves as a shortcut for models to learn and cannot generalize to brand-new classes. In contrast, models trained with only foreground "learn to compare" different objects—a desirable ability for reliable generalization to downstream few-shot learning tasks with out-of-domain classes.

**Training with background helps models to handle complex scenes.** When evaluating on $\mathcal{D}_v$-Ori, the models trained with original dataset $\mathcal{D}_B$-Ori are slightly better than those with foreground dataset $\mathcal{D}_B$-FG. We attribute this to a sort of domain shift: models trained with $\mathcal{D}_B$-FG never meet images with complex background and do not know how to handle it. In Appendix D.1 we further verify the assertion by showing evaluation accuracy of each class under the above two training situations. Note that since we apply random crop augmentation at training, domain shift does not exist if the models are instead trained on $\mathcal{D}_B$-Ori and evaluated on $\mathcal{D}_v$-FG.

**Simple fusion sampling combines advantages of both sides.** One may wish to cut off shortcut learning of background while maintaining adaptability of model to complex scenes. A simple solution may be fusion sampling: given an image as input, choose its foreground version with probability $p$, and its original version with probability $1 - p$. We simply set $p$ equal to $0.5$. We denote the dataset using this sampling strategy as $\mathcal{D}_B$-Fuse. As observed in Fig. 2(a), models trained this way indeed combine advantages of both sides: achieving relatively good performance on both $\mathcal{D}_v$-Ori and $\mathcal{D}_v$-FG. In Appendix C, we compare the training curves of PN trained on three versions of datasets to further investigate the effectiveness of fusion sampling.

The above analysis provides new inspiration for how to improve FSL further: (1) Fusion sampling of foreground and original images could be applied to training. (2) Since background information disturbs evaluation, it is needed to focus on foreground objects or assign image patches, that are more likely to be foreground, a larger weight for classification. Therefore, a foreground object identification mechanism is required at both training (for fusion sampling) and evaluation.

## 3.2 Contrastive Learning is Good at Identifying Objects

In this subsection, we reveal the potential of contrastive learning in identifying foreground objects, which we will use later for foreground extraction. Given one transformed view of one image, contrastive learning tends to distinguish another transformed view of that same image from thousands of views of other images. A more detailed introduction of contrastive learning is given in Appendix B. The two augmented views of the same image always cover the same object, but probably with different parts, sizes and color. To discriminate two augmented patches from thousands of other image patches, the model has to learn to identify the key discriminative information of the object under varying environment. In this manner, semantic relations among crops of images are explicitly modeled, thereby clustering semantically similar contents automatically. The features of different images are pushed away, while those of similar objects in different images are pulled closer. Thus it is reasonable to speculate that contrastive learning may enable models with better identification of centered foreground object.

To verify this, we train CC and contrastive learning models on the whole training set of *mini*ImageNet ($\mathcal{D}_B$-Full) and compare their accuracy on $\mathcal{D}_v$-Ori and $\mathcal{D}_v$-FG. The contrastive learning method we use is Exemplar [68], a modified version of MoCo [17]. Fig 2(b) shows that, while the evaluation accuracy of Exemplar on $\mathcal{D}_v$-Ori is slightly worse than that of CC, Exemplar performs much better when only foreground of images are given at evaluation, affirming that contrastive learning indeed has a better discriminative ability of single centered object. In Appendix D.2, we provide a more in-depth analysis of why contrastive learning has such properties and infer that the shape bias and viewpoint invariance may play an important role.

# 4 Rectifying the Shortcut Learning of Background

Given the analysis in the previous section, we wish to focus more on image foreground both at training and evaluation. Inspired by how humans recognise foreground objects, we propose COSOC, a framework ultilizing contrastive learning to draw the model's attention to the foreground objects of images.

## 4.1 Clustering-based Object Seeker (COS) with Fusion Sampling for Training

Since contrastive learning is good at discriminating foreground objects, we utilize it to extract foreground objects before training. The first step is to pre-train a backbone $f_\theta(\cdot)$ on the training set $\mathcal{D}_B$ using Exemplar [68]. Then a clustering-based algorithm is used to extract "objects" identified by the pre-trained model. The basic idea is that features of foreground objects in images within one class extracted by contrastive learning models are similar, thereby can be identified via a clustering algorithm; see a simple example in Fig. 3. All images within the $i$-th class in $\mathcal{D}_B$ form a set $\{\mathbf{x}_n^i\}_{n=1}^N$. For clarity, we omit the class index $i$ in the following descriptions. The scheme of seeking foreground objects in one class is detailed as follows:

1) For each image $\mathbf{x}_n$, we randomly crop it $L$ times to obtain $L$ image patches $\{\mathbf{p}_{n,m}\}_{m=1}^L$. Each image patch $\mathbf{p}_{n,m}$ is then passed through the pre-trained model $f_\theta$ and we get a normalized feature vector $\mathbf{v}_{n,m} = \frac{f_\theta(\mathbf{p}_{n,m})}{||f_\theta(\mathbf{p}_{n,m})||_2} \in \mathbb{R}^d$.

2) We run a clustering algorithm $\mathcal{A}$ on all features vectors of the class and obtain $H$ clusters $\{\mathbf{z}_j\}_{j=1}^H = \mathcal{A}(\{\mathbf{v}_{n,m}\}_{n,m=1}^{N,L})$, where $\mathbf{z}_j$ is the feature centroid of the $j$-th cluster.

3) We say an image $\mathbf{x}_n \in \mathbf{z}_j$, if there exists $k \in [L]$ s.t. $\mathbf{v}_{n,k} \in \mathbf{z}_j$, where $[L] = \{1, 2, \ldots, L\}$. Let $l(\mathbf{z}_j) = \frac{\#\{\mathbf{x}|\mathbf{x} \in \mathbf{z}_j\}}{N}$ be the proportion of images in the class that belong to $\mathbf{z}_j$. If $l(\mathbf{z}_j)$ is small, then the cluster $\mathbf{z}_j$ is not representative for the whole class and is possibly background. Thus we remove

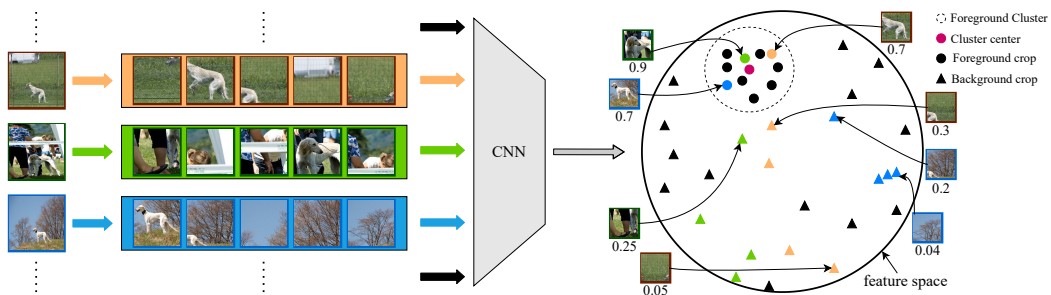

Figure 3: **Simplified schematic illustration of COS algorithm.** We show how we obtain foreground objects from three exemplified images. The value under each crop denotes its foreground score.

all the clusters $\mathbf{z}$ with $l(\mathbf{z}) < \gamma$, where $\gamma$ is a threshold that controls the generality of clusters. The remaining $h$ clusters $\{\mathbf{z}_j\}_{j=\alpha_1}^{\alpha_h}$ represent "objects" of the class that we are looking for.

4) The foreground score of image patch $p_{n,m}$ is defined as $s_{n,m} = 1 - \min_{j\in[h]}||\mathbf{v}_{n,m} - \mathbf{z}_{\alpha_j}||_2/\eta$, where $\eta = \max_{n,m} \min_{j\in[h_c]}||\mathbf{v}_{n,m} - \mathbf{z}_{\alpha_j}||_2$ is used to normalize the score into $[0,1]$. Then top-k scores of each image $\mathbf{x}_n$ are obtained as $\{s_{n,m}\}_{m=\beta_1}^{\beta_k} = \underset{m\in[L]}{\mathrm{Topk}}(s_{n,m})$. The corresponding patches $\{\mathbf{p}_{n,m}\}_{m=\beta_1}^{\beta_k}$ are seen as possible crops of the foreground object in image $\mathbf{x}_n$, and the foreground scores $\{s_{n,m}\}_{m=\beta_1}^{\beta_k}$ as the confidence. We then use it as prior knowledge to rectify the shortcut learning of background for FSL models.

The training strategy resembles fusion sampling introduced before. For an image $\mathbf{x}_n$, the probability that we choose the original version is $1 - \max_{i\in[k]} s_{n,\beta_i}$, and the probability of choosing $\mathbf{p}_{n,\beta_j}$ from top-k patches is $(s_{n,\beta_j}/\sum_{i\in[k]} s_{n,\beta_i}) \cdot \max_{i\in[k]} s_{n,\beta_i}$. Then we adjust the chosen image patch and make sure that the least area proportion to the original image keeps as a constant. We use this strategy to train a backbone $f_\theta(\cdot)$ using a FSL algorithm.

## 4.2 Few-shot Evaluation with Shared Object Concentrator (SOC)

As discussed before, if the foreground crop of the image is used at evaluation, the performance of FSL model will be boosted by a large margin, serving as an upper bound of the model performance. To approach this upper bound, we propose SOC algorithm to capture foreground objects by seeking shared contents among support images of the same class and query images.

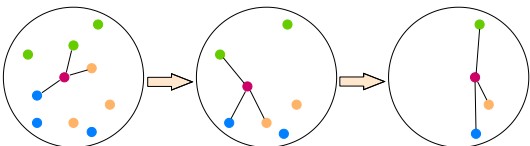

Figure 4: **The overall pipeline of step 1 in SOC.** Points in one color represent features of crops from one image. The red points are $\boldsymbol{\omega}_1, \boldsymbol{\omega}_2$ and $\boldsymbol{\omega}_3$.

**Step 1: Shared Content Searching within Each Class.** For each image $\mathbf{x}_k$ within one class $c$ from support set $\mathcal{S}_\tau$, we randomly crop it $V$ times and obtain corresponding candidates $\{\mathbf{p}_{k,n}\}_{n=1,..,V}$. Each patch $\mathbf{p}_{k,n}$ is individually sent to the learned backbone $f_\theta$ to obtain a normalized feature vector $\mathbf{v}_{k,n}$. Thus we have totally $K \times V$ feature vectors within a class $c$. Our goal is to obtain a feature vector $\boldsymbol{\omega}_1$ that contains maximal shared information of all images in class $c$. Ideally, $\boldsymbol{\omega}_1$ represents the centroid of the most similar $K$ image patches, each from one image, which can be formulated as

$$\boldsymbol{\omega}_1 = \frac{1}{K}\sum_{k=1}^{K} \mathbf{v}_{k,\lambda_{opt}(k)}, \tag{1}$$

$$\lambda_{opt} = \underset{\lambda\in[K]^{[V]}}{\arg\max} \sum_{1\leq i<j\leq K} \cos(\mathbf{v}_{i,\lambda(i)}, \mathbf{v}_{j,\lambda(j)}), \tag{2}$$

where $\cos(\cdot,\cdot)$ denotes cosine similarity and $[K]^{[V]}$ denotes the set of functions that take $[K]$ as domain and $[V]$ as range. While $\lambda_{opt}$ can be obtained by enumerating all possible combinations of image patches, the computation complexity of this brute-force method is $\mathcal{O}(V^K)$, which is

computation prohibitive when $V$ or $K$ is large. Thus when the computation is not affordable, we turn to use a simplified method that leverages iterative optimization. Instead of seeking for the closest image patches, we directly optimize $\boldsymbol{\omega}_1$ so that the sum of minimum distance to patches of each image is minimized, *i.e.*,

$$\boldsymbol{\omega}_1 = \underset{\boldsymbol{\omega} \in \mathcal{R}^d}{\arg\max} \sum_{k=1}^{K} \max_n [\cos(\boldsymbol{\omega}, \mathbf{v}_{k,n})], \tag{3}$$

which can be achieved by iterative optimization algorithms. We apply SGD in our experiments. After optimization, we remove the patch of each image that is most similar to $\boldsymbol{\omega}_1$, and obtain $K \times (V-1)$ feature vectors. Then we repeatedly implement the above optimization process until no features are left, as shown in Fig. 4. We eventually obtain $V$ sorted feature vectors $\{\boldsymbol{\omega}_n\}_{n=1}^{V}$, which we use to represent the class $c$. As for the case where shot $K = 1$, there is no shared inter-image information inside class, so similar to the handling in PN and DeepEMD [63], we just skip step 1 and use the original $V$ feature vectors.

**Step 2: Feature Matching for Concentrating on Foreground Object of Query Images.** Once the foreground class representations are identified, the next step is to use them to implicitly concentrate on foreground of query images by feature matching. For each image $\mathbf{x}$ in the query set $\mathcal{Q}_\tau$, we also randomly crop it for $V$ times and obtain $V$ candidate features $\{\boldsymbol{\mu}_n\}_{n=1}^{V}$. For each class $c$, we have $V$ sorted representative feature vectors $\{\boldsymbol{\omega}_n\}_{n=1}^{V}$ obtained in step 1. We then match the most similar patches between query features and class features, *i.e.*,

$$s_1 = \max_{1 \le i,j \le V} [\alpha^{j-1} \cos(\boldsymbol{\mu}_i, \boldsymbol{\omega}_j)], \tag{4}$$

where $\alpha \le 1$ is an importance factor. Thus the weight $\alpha^{j-1}$ decreases exponentially in index $n-1$, indicating a decreased belief of each vector representing foreground. Similarly, the two matched features are removed and the above process repeats until no features left. Finally, the score of $\mathbf{x}$ w.r.t. class $c$ is obtained as a weighted sum of all similarities, *i.e.*, $S_c = \sum_{n=1}^{V} \beta^{n-1} s_n$, where $\beta \le 1$ is another importance factor controlling the belief of each crop being foreground objects. In this way, features matched earlier—thus more likely to be foreground—will have higher contributions to the score. The predicted class of $\mathbf{x}$ is the one with the highest score.

# 5  Experiments

## 5.1  Experiment Setup

**Dataset.** We adopt two benchmark datasets which are the most representative in few-shot learning. The first is *mini*ImageNet [53], a small subset of ILSVRC-12 [44] that contains 600 images within each of the 100 categories. The categories are split into 64, 16, 20 classes for training, validation and evaluation, respectively. The second dataset, *tiered*ImageNet [41], is a much larger subset of ILSVRC-12 and is more challenging. It is constructed by choosing 34 super-classes with 608 categories. The super-classes are split into 20, 6, 8 super-classes which ensures separation between training and evaluation categories. The final dataset contains 351, 97, 160 classes for training, validation and evaluation, respectively. On both datasets, the input image size is 84 × 84 for fair comparison.

**Evaluation Protocols.** We follow the 5-way 5-shot (1-shot) FSL evaluation setting. Specifically, 2000 tasks, each contains 15 testing images and 5 (1) training images per class, are randomly sampled from the evaluation set $\mathcal{D}_v$ and the average classification accuracy is computed. This is repeated 5 times and the mean of the average accuracy with 95% confidence intervals is reported.

**Implementation Details.** The backbone we use throughout the article is ResNet-12, which is widely used in few-shot learning. We use Pytorch [38] to implement all our experiments on two NVIDIA 1080Ti GPUs. We train the model using SGD with cosine learning rate schedule without restart to reduce the number of hyperparameters (Which epochs to decay the learning rate). The initial learning rate for training Exemplar is 0.1, and for CC is 0.005. The batch size for Exemplar, CC are 256 and 128, respectively. For *mini*ImageNet, we train Exemplar for 150k iterations, and train CC for 6k iterations. For *tiered*ImageNet, we train Exemplar for approximately 900k iterations, and train CC for 120k iterations. We choose k-means [32] as the clustering algorithm for COS. The threshold $\gamma$ is set to 0.5, and top 3 out of 30 features are chosen per image at the training stage. At the evaluation stage, we crop each image 7 times. The importance factors $\alpha$ and $\beta$ are both set to 0.8.

Table 1: **Ablative study on *mini*ImageNet.** All models are trained on the full training set of *mini*ImageNet. Since the aim of SOC algorithm is to find foreground objects, it is unnecessary to evaluate SOC on the foreground dataset $\mathcal{D}_v$-FG. FT means finetuning from Exemplar used in COS.

| CC | FT | COS | SOC | $\mathcal{D}_v$-Ori | | $\mathcal{D}_v$-FG | |
| | | | | 1-shot | 5-shot | 1-shot | 5-shot |
|---|---|---|---|---|---|---|---|
| ✓ | | | | $62.67 \pm 0.32$ | $80.22 \pm 0.24$ | $66.69 \pm 0.32$ | $82.86 \pm 0.19$ |
| ✓ | | ✓ | | $64.76 \pm 0.13$ | $81.18 \pm 0.21$ | $\mathbf{71.13} \pm 0.36$ | $\mathbf{86.21} \pm 0.15$ |
| ✓ | ✓ | ✓ | | $65.05 \pm 0.06$ | $81.16 \pm 0.17$ | $\mathbf{71.36} \pm 0.30$ | $\mathbf{86.20} \pm 0.14$ |
| ✓ | ✓ | | ✓ | $64.41 \pm 0.22$ | $81.54 \pm 0.28$ | - | - |
| ✓ | ✓ | ✓ | ✓ | $\mathbf{69.29} \pm 0.12$ | $\mathbf{84.94} \pm 0.28$ | - | - |

Table 2: **Comparisons with baselines of foreground extractors using saliency detection algorithms on *mini*ImageNet.** For fair comparison, all models in the right column at evaluation use multi-cropping. GT means evaluating with ground truth foreground.

| Used for training | | | Used for evaluation | | |
| Method | 1-shot | 5-shot | Method | 1-shot | 5-shot |
|---|---|---|---|---|---|
| CC | $62.67 \pm 0.32$ | $80.22 \pm 0.24$ | COS | $67.23 \pm 0.35$ | $82.79 \pm 0.31$ |
| CC+RBD | $63.24 \pm 0.41$ | $80.45 \pm 0.37$ | COS+RBD | $67.03 \pm 0.52$ | $82.57 \pm 0.27$ |
| CC+MBD | $61.50 \pm 0.31$ | $79.12 \pm 0.32$ | COS+MBD | $62.98 \pm 0.45$ | $79.56 \pm 0.38$ |
| CC+FT | $62.71 \pm 0.11$ | $80.06 \pm 0.08$ | COS+FT | $64.74 \pm 0.28$ | $80.74 \pm 0.13$ |
| CC+COS | $\mathbf{64.76} \pm \mathbf{0.13}$ | $\mathbf{81.18} \pm \mathbf{0.21}$ | COSOC | $\mathbf{69.28} \pm \mathbf{0.49}$ | $\mathbf{85.16} \pm \mathbf{0.42}$ |
| - | - | - | COS+GT | $72.71 \pm 0.57$ | $87.43 \pm 0.36$ |

## 5.2 Model Analysis

In this subsection, we show the effectiveness of each component of our method. Tab. 1 shows the ablation study conducted on *mini*ImageNet.

**On the effect of finetuning.** Since a feature extractor is pre-trained using contrastive learning in COS, it may help accelerate convergence if we directly finetune from the pre-trained model instead of training from scratch. As shown in line 2-3 in Tab. 1, fintuning gives no improvement on the performance over training from scratch. Thus we adopt finetuning mainly for speeding up convergence ($5\times$ faster).

**Effectiveness of COS Algorithm.** As observed in Tab. 1, When COS is applied on CC, the performance is improved on both versions of datasets. In Fig. 5, we show the curves of training and validation error of CC during training with and without COS. Both models are trained from scratch and validated on the full *mini*ImageNet. We observe that CC sinks into overfitting: the training accuracy drops to zero, and validation accuracy stops improving before the end of the training. Meanwhile, the COS algorithm helps slow down convergence and prevent training accuracy from reaching zero. This makes validation accuracy comparable at first but higher at the end. Our COS algorithm weakens the "background shortcut" for learning, draws model's attention on foreground objects, and improves upon generalization.

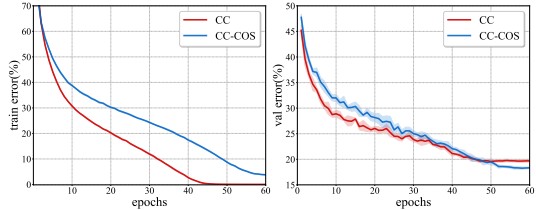

Figure 5: Comparison of training and validation curves between CC with and without COS.

**Effectiveness of SOC Algorithm.** The result in Tab. 1 shows that the SOC algorithm is the key to maximally exploit the potential of good object-discrimination ability. The performance even approaches the upper bound performance obtained by evaluating the model on the ground-truth foreground $\mathcal{D}_v$-FG. One potential unfairness in our SOC algorithm may lie in the use of multi-cropping, which could possibly lead to performance improvement for other approaches as well. We ablate this concern in Appendix G, as well as in the comparisons to other methods in the later subsections.

Table 3: **Comparisons with state-of-the-art models on *mini*ImageNet and *tiered*ImageNet.** The average **inductive** 5-way few-shot classification accuracies with 95 confidence interval are reported. * indicates methods evaluated using multi-cropping.

| Model | backbone | *mini*ImageNet | | *tiered*ImageNet | |
|---|---|---|---|---|---|
| | | 1-shot | 5-shot | 1-shot | 5-shot |
| MetaOptNet [22] | ResNet-12 | $62.64 \pm 0.82$ | $78.63 \pm 0.46$ | $65.99 \pm 0.72$ | $81.56 \pm 0.53$ |
| DC [28] | ResNet-12 | $62.53 \pm 0.19$ | $79.77 \pm 0.19$ | - | - |
| CTM [25] | ResNet-18 | $64.12 \pm 0.82$ | $80.51 \pm 0.13$ | $68.41 \pm 0.39$ | $84.28 \pm 1.73$ |
| CAM [19] | ResNet-12 | $63.85 \pm 0.48$ | $79.44 \pm 0.34$ | $69.89 \pm 0.51$ | $84.23 \pm 0.37$ |
| AFHN [26] | ResNet-18 | $62.38 \pm 0.72$ | $78.16 \pm 0.56$ | - | - |
| DSN [47] | ResNet-12 | $62.64 \pm 0.66$ | $78.83 \pm 0.45$ | $66.22 \pm 0.75$ | $82.79 \pm 0.48$ |
| AM3+TRAML [23] | ResNet-12 | $67.10 \pm 0.52$ | $79.54 \pm 0.60$ | - | - |
| Net-Cosine [29] | ResNet-12 | $63.85 \pm 0.81$ | $81.57 \pm 0.56$ | - | - |
| CA [2] | WRN-28-10 | $65.92 \pm 0.60$ | $82.85 \pm 0.55$ | $\mathbf{74.40} \pm \mathbf{0.68}$ | $86.61 \pm 0.59$ |
| MABAS [21] | ResNet-12 | $65.08 \pm 0.86$ | $82.70 \pm 0.54$ | - | - |
| ConsNet [59] | ResNet-12 | $64.89 \pm 0.23$ | $79.95 \pm 0.17$ | - | - |
| IEPT [66] | ResNet-12 | $67.05 \pm 0.44$ | $82.90 \pm 0.30$ | $72.24 \pm 0.50$ | $86.73 \pm 0.34$ |
| MELR [11] | ResNet-12 | $67.40 \pm 0.43$ | $83.40 \pm 0.28$ | $72.14 \pm 0.51$ | $87.01 \pm 0.35$ |
| IER-Distill [42] | ResNet-12 | $67.28 \pm 0.80$ | $84.78 \pm 0.52$ | $72.21 \pm 0.90$ | $87.08 \pm 0.58$ |
| LDAMF [57] | ResNet-12 | $67.76 \pm 0.46$ | $82.71 \pm 0.31$ | $71.89 \pm 0.52$ | $85.96 \pm 0.35$ |
| FRN [55] | ResNet-12 | $66.45 \pm 0.19$ | $82.83 \pm 0.13$ | $72.06 \pm 0.22$ | $86.89 \pm 0.14$ |
| Baseline* [7] | ResNet-12 | $63.83 \pm 0.67$ | $81.38 \pm 0.41$ | - | - |
| DeepEMD* [63] | ResNet-12 | $67.63 \pm 0.46$ | $83.47 \pm 0.61$ | $74.29 \pm 0.32$ | $86.98 \pm 0.60$ |
| RFS-Distill* [51] | ResNet-12 | $65.02 \pm 0.44$ | $82.04 \pm 0.38$ | $71.52 \pm 0.69$ | $86.03 \pm 0.49$ |
| FEAT* [60] | ResNet-12 | $68.03 \pm 0.38$ | $82.99 \pm 0.31$ | - | - |
| Meta-baseline* [8] | ResNet-12 | $65.31 \pm 0.51$ | $81.26 \pm 0.23$ | $68.62 \pm 0.27$ | $83.74 \pm 0.18$ |
| **COSOC* (ours)** | ResNet-12 | $\mathbf{69.28} \pm \mathbf{0.49}$ | $\mathbf{85.16} \pm \mathbf{0.42}$ | $73.57 \pm 0.43$ | $\mathbf{87.57} \pm \mathbf{0.10}$ |

Note that if we apply only the SOC algorithm on CC, the performance degrades. This indicates that COS and SOC are both necessary: COS provides the discrimination ability of foreground objects and SOC leverages it to maximally boost the performance.

## 5.3 Comparison to Saliency-based Foreground Extractors

There could be other possible ways of extracting foreground objects. A simple yet possibly strong baseline could be running saliency detection to extract the most salient region in an image, followed by cropping to obtain patches without background. We consider comparing with three classical unsupervised saliency methods—RBD [69], FT [1] and MBD [65]. The cropping threshold is specially tuned. For training, fusion sampling with probability 0.5 is used for unsupervised saliency methods. For evaluation, We replace the original images with crops obtained by unsupervised saliency methods directly for classification. Tab. 2 displays the comparisons of performance using different foreground extraction methods applied at training or evaluation. For fair comparison, all methods are trained from scratch, and all compared baselines are evaluated with multi-cropping (i.e. using the average of features obtained from multiple crops for classification)and tested on the same backbone (COS trained).

The results show that: (1) Our method performs consistently much better than the listed unsupervised saliency methods. (2) The performance of different unsupervised saliency methods varies. While RBD gives a small improvement, MBD and FT have negative effect on the performance. The performance severely depends on the effectiveness of unsupervised saliency methods, and is very sensitive to the cropping threshold. Intuitively speaking, saliency detection methods focus on noticeable objects in the image, and might fail when there is another irrelevant salient object in the image (e.g., a man is walking a dog. Dog is the label, but the man is of high saliency). On the contrary, our method focuses on shared objects across images in the same class, thereby avoiding this problem. In addition, our COS algorithm has the ability to dynamically assign foreground scores to different patches, which reduces the risk of overconfidence. One of our main contributions is paving a new way towards improving FSL by rectifying shortcut learning of background, which can be implemented using any effective methods. Given the upper bound with ground truth foreground, we believe there is room to improve and there can be other more effective approaches in the future.

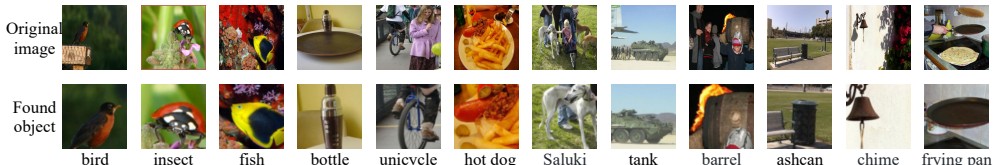

bird insect fish bottle unicycle hot dog Saluki tank barrel ashcan chime frying pan

Figure 6: **Examples of objects obtained with COS from the training set of *mini*ImageNet.** The first row shows the original images;the second row shows the picked patch with the highest foreground score.

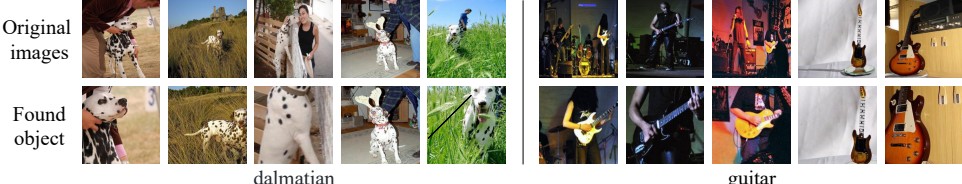

dalmatian                                              guitar

Figure 7: **Visualization examples of the SOC algorithm.** The first row displays 5 images that belong to dalmatian and guitar classes respectively from evaluation set of miniImageNet. The second row shows image patches that are picked up from the first round of SOC algorithm. Our method succesfully puts focus on the shared contents/foreground.

## 5.4 Comparison to State-of-the-Arts

Tab. 3 presents 5-way 1-shot and 5-shot classification results on *mini*ImageNet and *tiered*ImageNet. We compare with state-of-the-art few-shot learning methods. For fair comparison, we reimplement some methods, and evaluate them with multi-cropping. See Appendix G for a detailed study on the influence of multi-cropping. Our method achieves state-of-the-art performance under all settings except for 1-shot task on *tiered*ImageNet, on which the performance of our method is slightly worse than CA, which uses WRN-28-10, a deeper backbone, as the feature extractor.

## 5.5 Visualization

Fig. 6 and 7 display visualization examples of the COS and SOC algorithms. See more examples in Appendix H. Thanks to the well-designed mechanism of capturing shared inter-image information, the COS and SOC algorithms are capable of locating foreground patches embodied in complicated, multi-object scenery.

## 6 Conclusion

Few-shot image classification benefits from increasingly more complex network and algorithm design, but little attention has been focused on image itself. In this paper, we reveal that image background serves as a source of harmful knowledge that few-shot learning models easily absorb in. This problem is tackled by our COSOC framework that can draw the model's attention to image foreground at both training and evaluation. Our method is only one possible solution, and future work may include exploring the potential of unsupervised segmentation or detection algorithms which may be a more reliable alternative of random cropping, or looking for a completely different but better algorithm customized for foreground extraction.

## Acknowledgments and Disclosure of Funding

Special thanks to Qi Yong, who gives indispensable support on the spirit of this paper. We also thank Junran Peng for his help and fruitful discussions. This paper was partially supported by the National Key Research and Development Program of China (No. 2018AAA0100204), and a key program of fundamental research from Shenzhen Science and Technology Innovation Commission (No. JCYJ20200109113403826).

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
