# Appendix for Rectifying the Shortcut Learning of Background for Few-Shot Learning

## A Details of Section 3

**Dataset Construction.** We construct a subset $\mathcal{D} = (\mathcal{D}_B, \mathcal{D}_v)$ of *mini*ImageNet $\mathcal{D}$-Full $= (\mathcal{D}_B$-Full, $\mathcal{D}_v$-Full). $\mathcal{D}_B$ is created by randomly picking 100 out of 600 images from the first 27 categories of $\mathcal{D}_B$-Full; And $\mathcal{D}_v$ is created by randomly picking 40 out of 600 images from all categories of $\mathcal{D}_v$-Full. We then crop each image in $\mathcal{D}$ such that the foreground object is tightly bounded. Some examples are displayed in Fig. 1.

**Cosine Classifier (CC) and Prototypical Network (PN).** In CC [3], the feature extractor $f_\theta$ is trained together with a cosine-similarity based classifier under standard supervised way. The loss can be formally described as

$$\mathcal{L}^{\text{CC}} = -\mathbb{E}_{(x,y)\sim D_B}\Big[\log \frac{e^{\cos(f_\theta(x), w_y)}}{\sum_{i=1}^{C} e^{\cos(f_\theta(x), w_i)}}\Big], \tag{1}$$

where $C$ denotes the number of classes in $\mathcal{D}_B$, $\cos(\cdot, \cdot)$ denotes cosine similarity and $w_i \in \mathbb{R}^d$ denotes the learnable prototype for class $i$. To solve a following downstream few-shot classification task $(\mathcal{S}_\tau, \mathcal{Q}_\tau) \in \mathcal{T}$, CC adopts a non-parametric metric-based algorithm. Specifically, all images in $(\mathcal{S}_\tau, \mathcal{Q}_\tau)$ are mapped into features by the trained feature extractor $f_\theta$. Then all features from the same class $c$ in $\mathcal{S}_\tau$ are averaged to form a prototype $p_c = \frac{1}{K}\sum_{(x,y)\in\mathcal{S}_\tau} \mathbb{1}_{[y=c]} f_\theta(x)$. Cosine similarity between query image and each prototype is then calculated to obtain score w.r.t. the corresponding class. In summary, the score for a test image $x_q$ w.r.t. class $c$ can be written as

$$S_c(x_q; \mathcal{S}_\tau) = \log \frac{e^{\cos(f_\theta(x_q), p_c)}}{\sum_{i=1}^{N} e^{\cos(f_\theta(x_q), p_i)}}, \tag{2}$$

and the predicted class for $x_q$ is the one with the highest score.

The difference between PN and CC is only at the training stage. PN follows meta-learning/episodic paradigm, in which a pseudo $N$-way $K$-shot classication task $(\mathcal{S}_t, \mathcal{Q}_t)$ is sampled from $\mathcal{D}_B$ during each iteration $t$ and is solved using the same algorithm as (2). The loss at iteration $t$ is the average prediction loss of all test images and can be described as

$$\mathcal{L}_t^{\text{PN}} = -\frac{1}{|\mathcal{Q}_t|} \sum_{(x,y)\in\mathcal{Q}_t} S_y(x; \mathcal{S}_t). \tag{3}$$

**Implementation Details in Sec. 3.** For all experiments in Sec. 3, we train CC and PN with ResNet-12 for 60 epochs. The initial learning rate is 0.1 with cosine decay schedule without restart. Random crop is used as data augmentation. The batch size for CC is 128 and for PN is 4.

## B Contrastive Learning

Contrastive learning tends to maximize the agreement between transformed views of the same image and minimize the agreement between transformed views of different images. Specifically, Let $f_\phi(\cdot)$

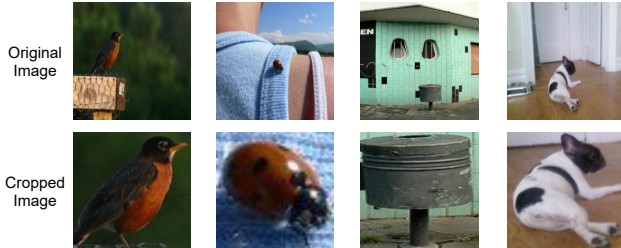

Original Image

Cropped Image

Figure 1: Examples of images of constructed datasets $\mathcal{D}$. The first row shows images in $\mathcal{D}_B$ which are original images of *mini*ImageNet; and the second row illustrates corresponding cropped versions in $\mathcal{D}_v$ in which only foreground objects are remained.

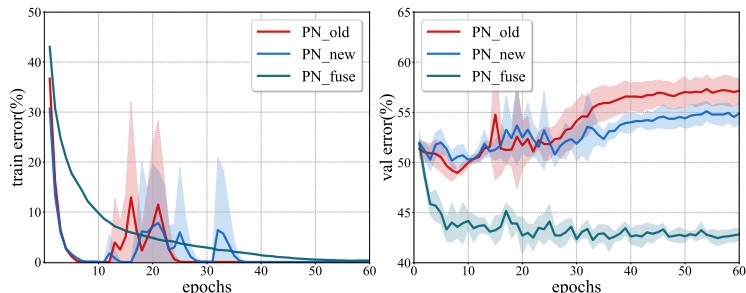

Figure 2: Comparison of training and validation curves of PN trained under three different settings.

be a convolutional neural network with output feature space $\mathbb{R}^d$. Two augmented image patches from one image $x$ are mapped by $f_\phi(\cdot)$, producing one query feature $\mathbf{q}$, and one key feature $\mathbf{k}$. Additionally, a queue containing thousands of negative features $\{v_n\}_{n=1}^Q$ is produced using patches of other images. This queue can either be generated online using all images in the current batch [1] or offline using stored features from last few epochs [4]. Given $q$, contrastive learning aims to identify $k$ in thousands of features $\{v_n\}_{n=1}^Q$, and can be formulated as:

$$\mathcal{L}(\mathbf{q}, \mathbf{k}, \{v_n\}) = -\log \frac{e^{\text{sim}(\mathbf{q},\mathbf{k})/\tau}}{e^{\text{sim}(\mathbf{q},\mathbf{k})/\tau} + \sum_{j=1}^Q e^{\text{sim}(\mathbf{q},v_j)/\tau}}, \tag{4}$$

Where $\tau$ denotes a temperature parameter, $\text{sim}(\cdot, \cdot)$ a similarity measure. In Exemplar [5], all samples in $\{v_n\}_{n=1}^Q$ that belong to the same class as $\mathbf{q}$ are removed in order to *"preserve the unique information of each positive instance while utilizing the label information in a weak manner"*.

## C  Shortcut Learning in PN

Fig. 2 shows training and validation curves of PN trained on $\mathcal{D}_B$-Ori, $\mathcal{D}_B$-FG and $\mathcal{D}_B$-Fuse. It can be observed that the training errors of models trained on $\mathcal{D}_B$-Ori and $\mathcal{D}_B$-FG both decrease to zero within 10 epochs. However, the validation error does not decrease to a relatively low value and remains high after convergence, reflecting severe overfitting phenomenon. On the contrary, PN with fusion samping converges much slower with a relatively lower validation error at the end. Apparently, shortcuts for PN on both $\mathcal{D}_B$-Ori and $\mathcal{D}_B$-FG exist and are suppressed by fusion sampling. In our paper we have showed that the shortcuts for dataset $\mathcal{D}_B$-Ori may be the statistical correlations between background and label and can be relieved by foreground concentration. However for dataset $\mathcal{D}_B$-FG the shortcut is not clear, and we speculate that appropriate amount of background information injects some noisy signals into the optimizatiton process which can help the model escape from local minima. We leave it for future work to further exploration.

## D  Comparisons of Class-wise Evaluation Performance

Common few-shot evaluation focuses on the average performance of the whole evaluation set, which can not tell a method is why and in what aspect better than another one. To this end, we propose a

Table 1: Comparisons of class-wise evaluation performance. The first row shows the training sets of which we compare different models. The second row shows the dataset we evaluate on. Each score denotes the difference of average accuracy of one class, e.g. a vs. b: (performance of a) - (performance of b).

| $\mathcal{D}_B$-FG vs. $\mathcal{D}_B$-Ori $\mathcal{D}_v$-Ori | | $\mathcal{D}_B$-FG vs. $\mathcal{D}_B$-Ori $\mathcal{D}_v$-FG | | $\mathcal{D}_B$-Full: Exemplar vs CC $\mathcal{D}_v$-FG | |
|---|---|---|---|---|---|
| class | score | class | score | class | score |
| trifle | +3.81 | theater curtain | +7.39 | electric guitar | +17.28 |
| theater curtain | +3.47 | mixing bow | +7.04 | vase | +10.64 |
| mixing bowl | +1.61 | trifle | +4.33 | ant | +8.88 |
| vase | +1.13 | vase | +3.84 | nematode | +7.72 |
| nematode | +0.12 | ant | +3.62 | cuirass | +4.63 |
| school bus | -0.52 | scoreboard | +2.92 | mixing bowl | +4.30 |
| electric guitar | -0.87 | crate | +1.18 | theater curtain | +3.35 |
| black-footed ferret | -0.91 | nematode | +0.93 | bookshop | +2.27 |
| scoreboard | -1.55 | lion | +0.83 | crate | +1.64 |
| bookshop | -1.60 | electric guitar | +0.61 | lion | +1.46 |
| lion | -2.04 | hourglass | -0.57 | African hunting dog | +1.42 |
| hourglass | -3.12 | black-footed ferret | -0.69 | trifle | +1.13 |
| African hunting dog | -3.99 | school bus | -0.86 | scoreboard | +1.06 |
| cuirass | -4.05 | king crab | -1.95 | schoolbus | +0.73 |
| king crab | -4.44 | bookshop | -2.25 | hourglass | -0.94 |
| crate | -5.32 | cuirass | -2.54 | dalmatian | -1.66 |
| ant | -5.50 | golden retriever | -3.18 | malamute | -2.39 |
| dalmatian | -9.71 | African hunting dog | -3.84 | king crab | -2.48 |
| golden retriever | -10.27 | dalmatian | -3.90 | golden retriever | -3.13 |
| malamute | -12.00 | malamute | -5.72 | black-footed ferret | -5.81 |

more fine-grained class-wise evaluation protocol which displays average few-shot performance per class instead of single average performance.

We first visualize some images from each class of $\mathcal{D}_v$-Ori in Fig. 3. The classes are sorted by Signal-to-Full (SNF) ratio, which is the average ratio of foreground area over original area in each class. For instance, the class with highest SNF is *bookshop*. The images within this class always display a whole indoor scene, which can be almost fully recognised as foreground. In contrast, images from the class *ant* always contain large parts of background which are irrelevant with the category semantics, thus have low SNF. Although the SNF may not reflect the true complexity of background, we use it as an indicator and hope we could obtain some insights from the analysis.

### D.1 Domain Shift

We first analyse the phenomenon of domain shift of few-shot models trained on $\mathcal{D}_B$-FG and evaluated on $\mathcal{D}_B$-FG. The first column in Tab. 1 displays class-wise performance difference between CC trained on $\mathcal{D}_B$-FG and $\mathcal{D}_B$-Ori. It can be seen that the worst-performance classes of model trained on $\mathcal{D}_B$-FG are those with low SNF and complex background. This indicates that the model trained on $\mathcal{D}_B$-FG fails to recognise objects taking up small space because they have never met such images during training.

### D.2 Shape Bias and View-Point Invariance of Contrastive Learning

The third column of Tab. 1 shows the class-wise performance difference between Exemplar and CC evaluated on $\mathcal{D}_v$-FG. We at first take a look at classes on which contrastive learning performs much better than CC: *electric guitar*, *vase*, *ant*, *nematode*, *cuirass* and *mixing bowl*. One observation is that the objects of each of these classes look similar in shape. Geirhos et al. [2] point out that CNNs are strongly biased towards recognising textures rather than shapes, which is different from what humans do and is harmful for some downstream tasks. Thus we speculate that one of the reasons that contrastive learning is better than supervised models in some aspects is that contrastive learning

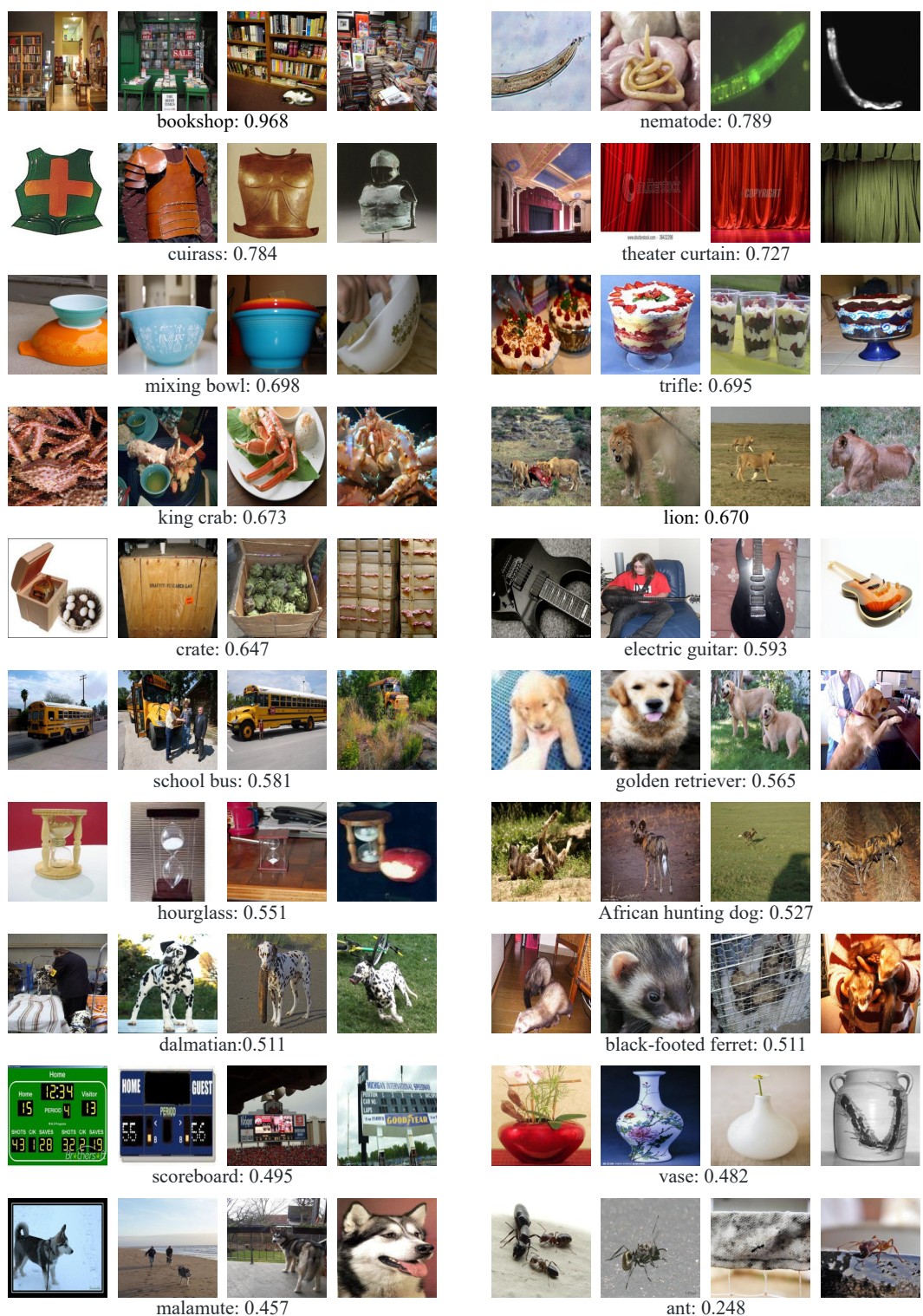

Figure 3: Illustrative examples of images in $\mathcal{D}_v$-Ori. The number under each class of images denotes Signal-to-Full ratio (SNF) ratio which is the average ratio of foreground area over original area in each class. Higher SNF approximately means less noise inside images.

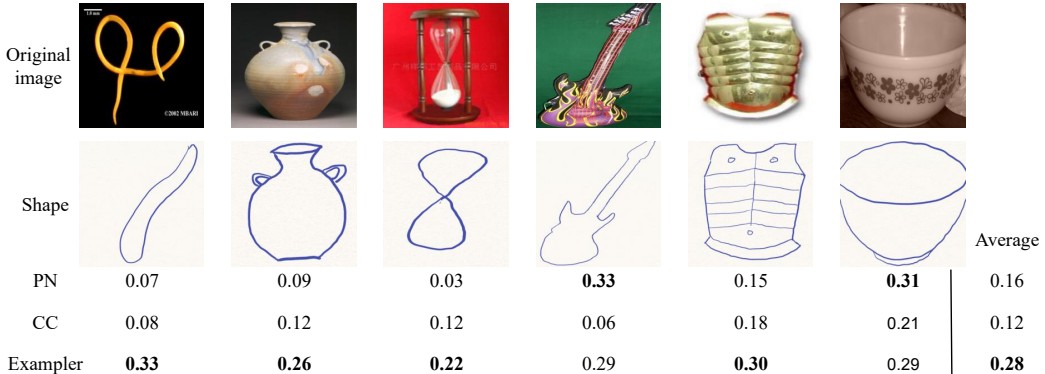

| | | | | | | Average |
|---|---|---|---|---|---|---|
| PN | 0.07 | 0.09 | 0.03 | **0.33** | 0.15 | **0.31** | 0.16 |
| CC | 0.08 | 0.12 | 0.12 | 0.06 | 0.18 | 0.21 | 0.12 |
| Exampler | **0.33** | **0.26** | **0.22** | 0.29 | **0.30** | 0.29 | **0.28** |

Figure 4: Shape similarity test. Each number denotes the feature similarity between the above image and its shape using correponding trained feature extractor.

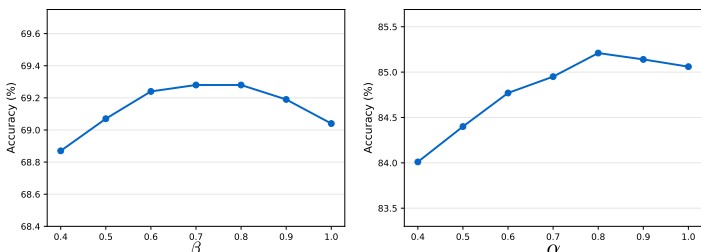

Figure 5: The effect of different values of $\beta$ and $\alpha$. The left figure shows the 5-way 1-shot accuracies, while the right figure shows the 5-way 5-shot accuracies with $\beta$ fixed as $0.8$.

prefers shape information more to recognising objects. To simply verify this, we hand draw shapes of some examples from the evaluation dataset; see Fig. 4. Then we calculate the similarity between features of original images and the shape image using different feature extractors. The results are shown in Fig. 4. As we can see, Exemplar recognises objects based on shape information more than the other two supervised methods. This is a conjecture more than a assertion. We leave it for future work to explore the shape bias of contrastive learning more deeply.

Next, let's have a look on the classes on which contrastive learning performs relatively poor: *black-footed ferret*, *golden retriever*, *king crab* and *malamute*. It can be noticed that these classes all refer to animals that have different shapes under different view points. For example, dogs from the front and dogs from the side look totally different. The supervised loss pulls all views of one kind of animals closer, therefore enabling the model with the knowledge of dicriminating objects from different view points. On the contrary, contrastive learning pushes different images away, but only pulls patches of the *same* one image which has the same view point, thus has no prior of view point invariance. This suggests that contrastive learning can be further improved if view point invariance is injected into the learning process.

### D.3 The Similarity between training Supervised Models with Foreground and training Models with Contrastive Learning

The second column and the third column of Tab. 1 are somehow similar, indicating that supervised models learned with foreground and learned with contrastive learning learn similar patterns of images. However, there are some classes that have distinct performance. For instance, the performance difference of contrastive learning on class *electric guitar* over CC is much higher than that of CC with $\mathcal{D}_B$-FG over $\mathcal{D}_B$-Ori. It is interesting to investigate what makes the difference between the representations learned by contrastive learning and supervised learning.

Table 2: 5-way few-shot performance of CC and PN with different variants of training and evaluation datasets.

| Model | training set | $\mathcal{D}_v$-Ori | | $\mathcal{D}_v$-FG | |
|---|---|---|---|---|---|
| | | 1-shot | 5-shot | 1-shot | 5-shot |
| CC | $\mathcal{D}_B$-Ori | $45.29 \pm 0.27$ | $62.73 \pm 0.36$ | $49.03 \pm 0.28$ | $66.75 \pm 0.15$ |
| | $\mathcal{D}_B$-FG | $44.84 \pm 0.20$ | $60.85 \pm 0.32$ | $\mathbf{52.22} \pm 0.35$ | $68.65 \pm 0.22$ |
| | $\mathcal{D}_B$-Fuse | $\mathbf{46.02} \pm 0.18$ | $\mathbf{62.91} \pm 0.40$ | $51.87 \pm 0.39$ | $\mathbf{68.98} \pm 0.22$ |
| PN | $\mathcal{D}_B$-Ori | $40.57 \pm 0.32$ | $52.74 \pm 0.11$ | $44.24 \pm 0.45$ | $56.75 \pm 0.34$ |
| | $\mathcal{D}_B$-FG | $40.25 \pm 0.36$ | $53.25 \pm 0.33$ | $46.93 \pm 0.50$ | $61.16 \pm 0.35$ |
| | $\mathcal{D}_B$-Fuse | $\mathbf{45.25} \pm 0.44$ | $\mathbf{59.23} \pm 0.28$ | $\mathbf{50.72} \pm 0.43$ | $\mathbf{64.96} \pm 0.20$ |

Table 3: Comparisons of 5-way few-shot performance of CC and Exemplar trained on the full *mini*ImageNet and evaluated on two versions of evaluation datasets.

| Model | $\mathcal{D}_v$-Ori | | $\mathcal{D}_v$-FG | |
|---|---|---|---|---|
| | 1-shot | 5-shot | 1-shot | 5-shot |
| CC | $\mathbf{62.67} \pm 0.32$ | $\mathbf{80.22} \pm 0.23$ | $66.69 \pm 0.32$ | $82.86 \pm 0.20$ |
| Exemplar | $61.14 \pm 0.14$ | $78.13 \pm 0.23$ | $\mathbf{70.14} \pm 0.12$ | $\mathbf{85.12} \pm 0.21$ |

# E  Additional Ablative Studies

In Fig. 5, we show how different values of $\beta$ and $\alpha$ influence the performance of our model. $\beta$ and $\alpha$ serve as importance factors in SOC, that express the belief of our firstly obtained foreground objects. As we can see, the performance of our model suffers from either excessively firm (small values) or weak (high values) belief. As $\alpha$ and $\beta$ approach zero, it puts more attention on the first few detected objects, leading to increasing risk of wrong matchings of foreground objects; as $\alpha$ and $\beta$ approach one, all weights of features tend to be the same, losing more emphasis on foreground objects.

# F  Detailed Performance in Sec. 3

We show detailed performance (both 1-shot and 5-shot) in Tab. 2 and Tab. 3. From the tables, we can see that 5-way 1-shot performance follows the same trend as 5-way 5-shot performance discussed in the main article.

# G  The Influence of Multi-cropping

For fair comparison and to better clarify the influence of our SOC algorithm, we include additional experiments about the influence of multi-cropping. We implemented several Few-Shot Learning methods using multi-cropping during evaluation. Specifically, for all methods except DeepEMD, we average the feature vectors of 7 crops and use the resulted averaged feature for classification. For DeepEMD, we notice that they also report performance using multi-cropping during the evaluation

Table 4: The influence of multi-cropping on *mini*ImageNet.

| Method | 1-shot (no MC $\rightarrow$ MC) | 5-shot (no MC $\rightarrow$ MC) |
|---|---|---|
| PN | 60.19→63.97 | 75.50→78.90 |
| Baseline | 60.93→63.83 | 78.46→81.38 |
| CC | 62.67→64.41 | 80.22→82.74 |
| Meta-baseline | 62.65→65.31 | 79.10→81.26 |
| RFS-distill | 63.00→65.02 | 79.63→82.04 |
| FEAT | 66.45→68.03 | 81.94→82.99 |
| DeepEMD | 66.61→67.63 | 82.02→83.47 |
| S2M2_R | 64.93→66.97 | 83.18→84.16 |
| COS | 65.05→67.23 | 81.16→82.79 |
| COSOC | 69.28(with MC) | 85.16(with MC) |
| COS+groundtruth | 71.36→72.71 | 86.20→87.43 |

stage, thus we follow the method in the original paper. We report the results in Tab. 4. As a reference of upper bound, we have also included the performance of using the ground truth foreground. We denote multi-cropping as MC. The results show that multi-cropping can improve FSL models by 1-3 points, and the improvement tends to be marginal when the baseline performance becomes higher. Moreover, the improvement is smaller in 5-shot settings.

## H  Additional Visualization Results

In Fig. 6-9 we display more visualization results of COS algorithm on four classes from the training set of *mini*ImageNet. For each image, we show the top 3 out of 30 crops with the highest foreground scores. From the visualization results, we can conclude that: (1) our COS algorithm can reliably extract foreground regions from images, even if the foreground objects are very small or backgrounds are extremely noisy. (2) When there is an object in the image which is similar with the foreground object but comes from a distinct class, our COS algorithm can accurately distinguish them and focus on the right one, e.g. the last group of pictures in Fig. 7. (3) When multiple instances of foreground object exist in one picture, our COS algorithm can capture them simultaneously, distributing them in different crops, e.g. last few groups in Fig. 6. Fig. 10 shows additional visulization results of SOC algorithms. Each small group of images display one 5-shot example from one class of the evaluation set of *mini*ImageNet. Similar observations are presented, consistent with those in the main article.

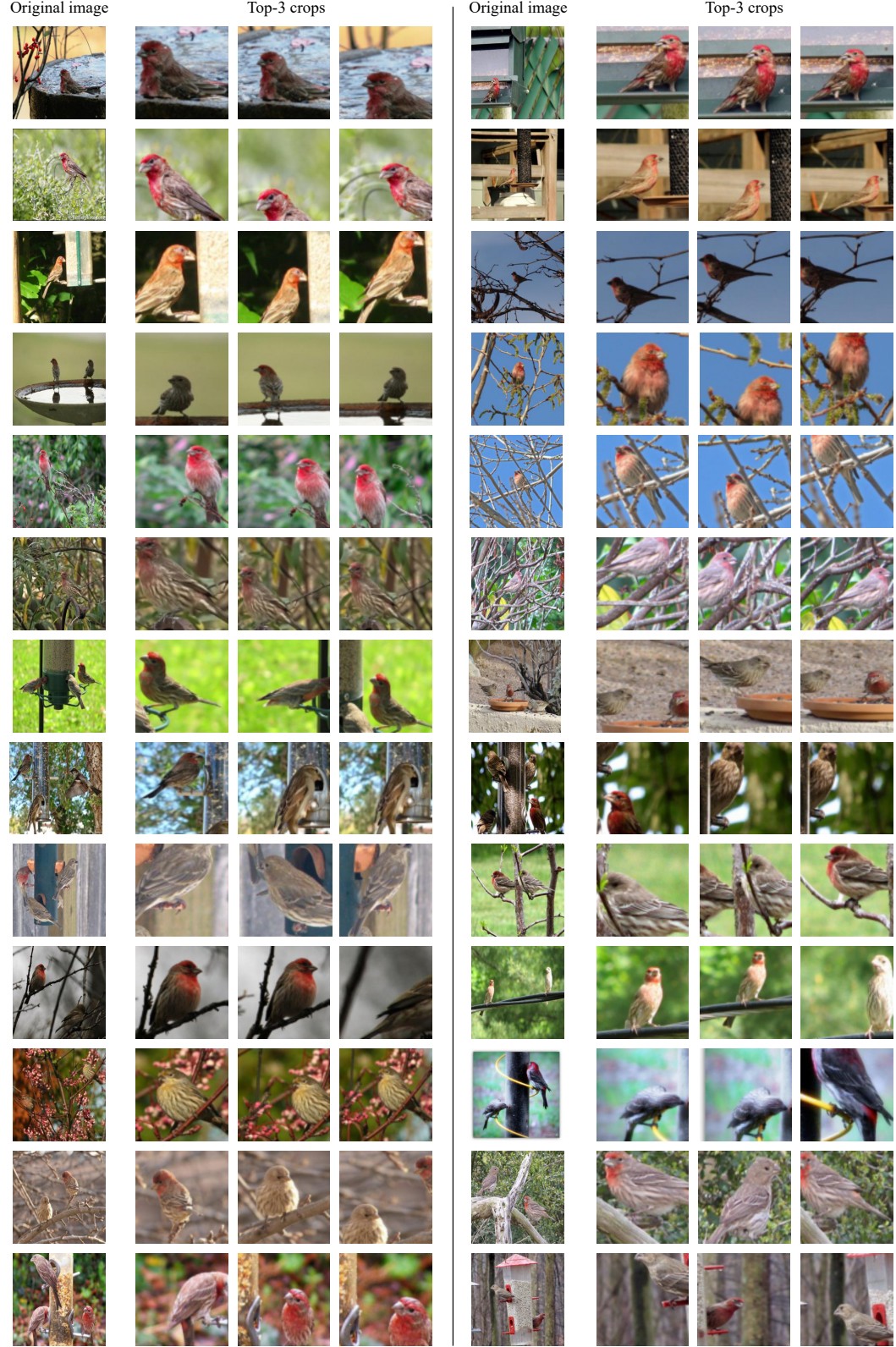

Original image    Top-3 crops                    Original image    Top-3 crops

Figure 6: Visulization results of COS algorithm on class ***house finch***.

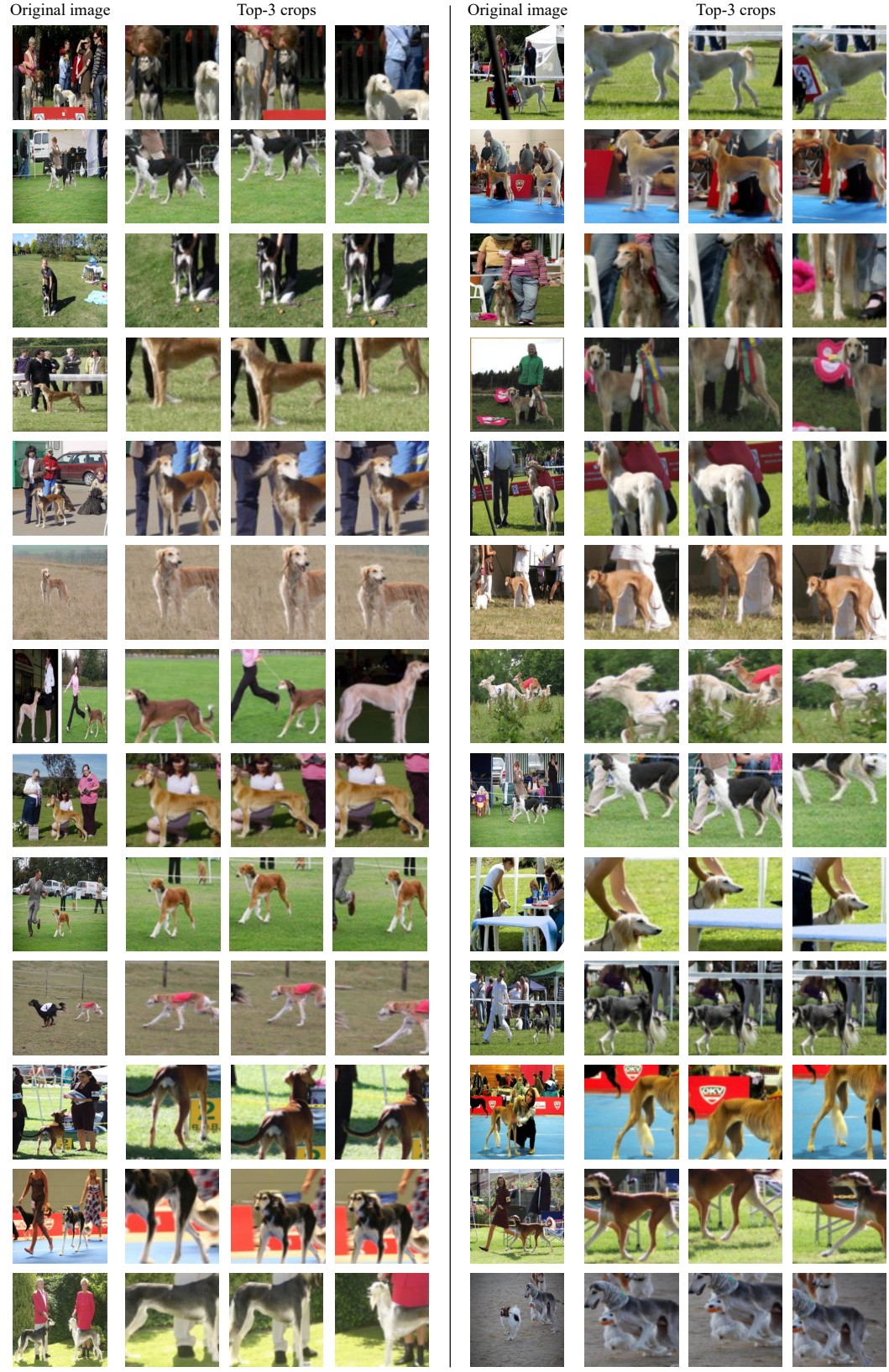

Figure 7: Visulization results of COS algorithm on class *Saluki*.

Original image          Top-3 crops          Original image          Top-3 crops

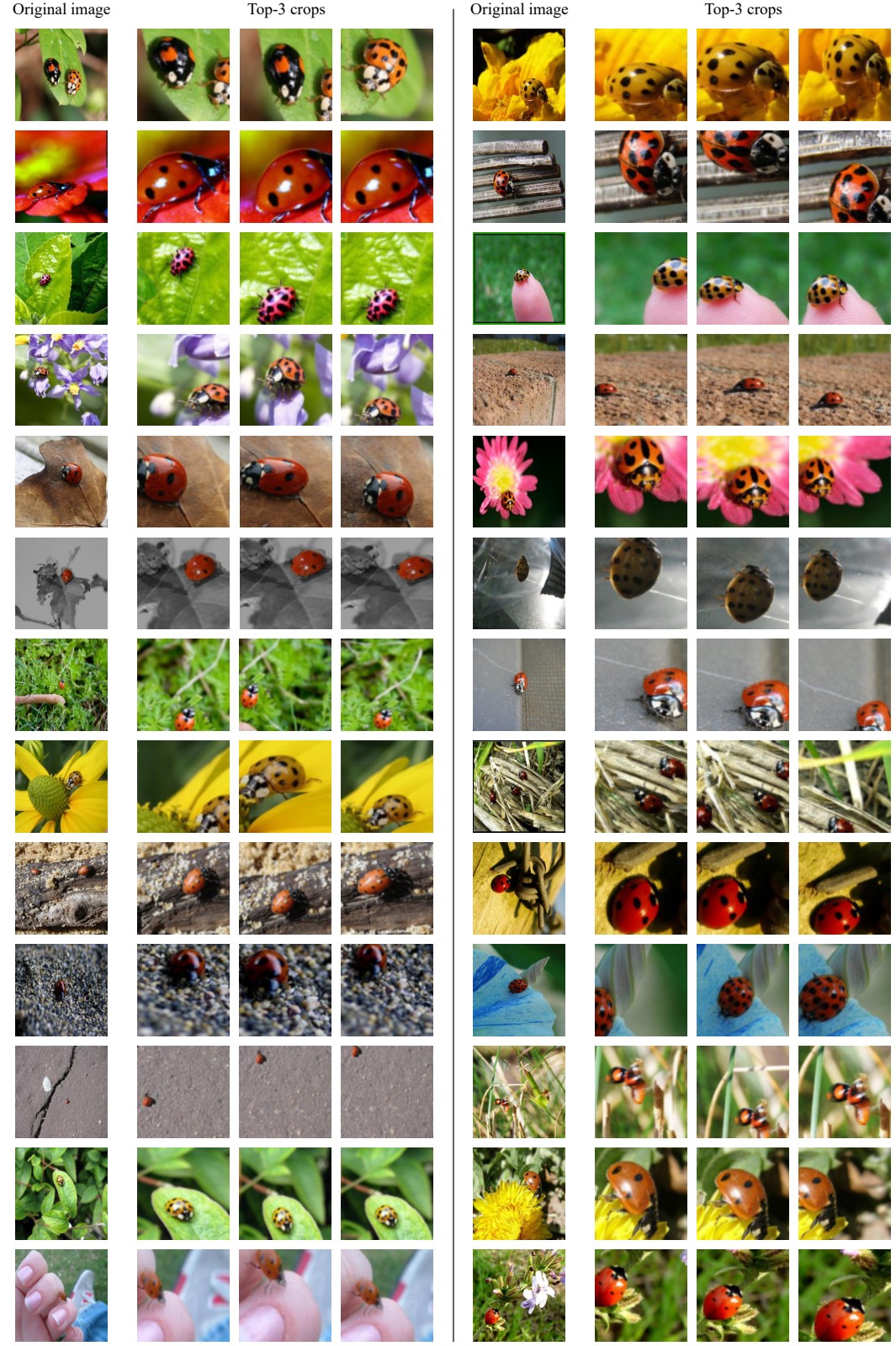

Figure 8: Visulization results of COS algorithm on class ***ladybug***.

Original image          Top-3 crops          Original image          Top-3 crops

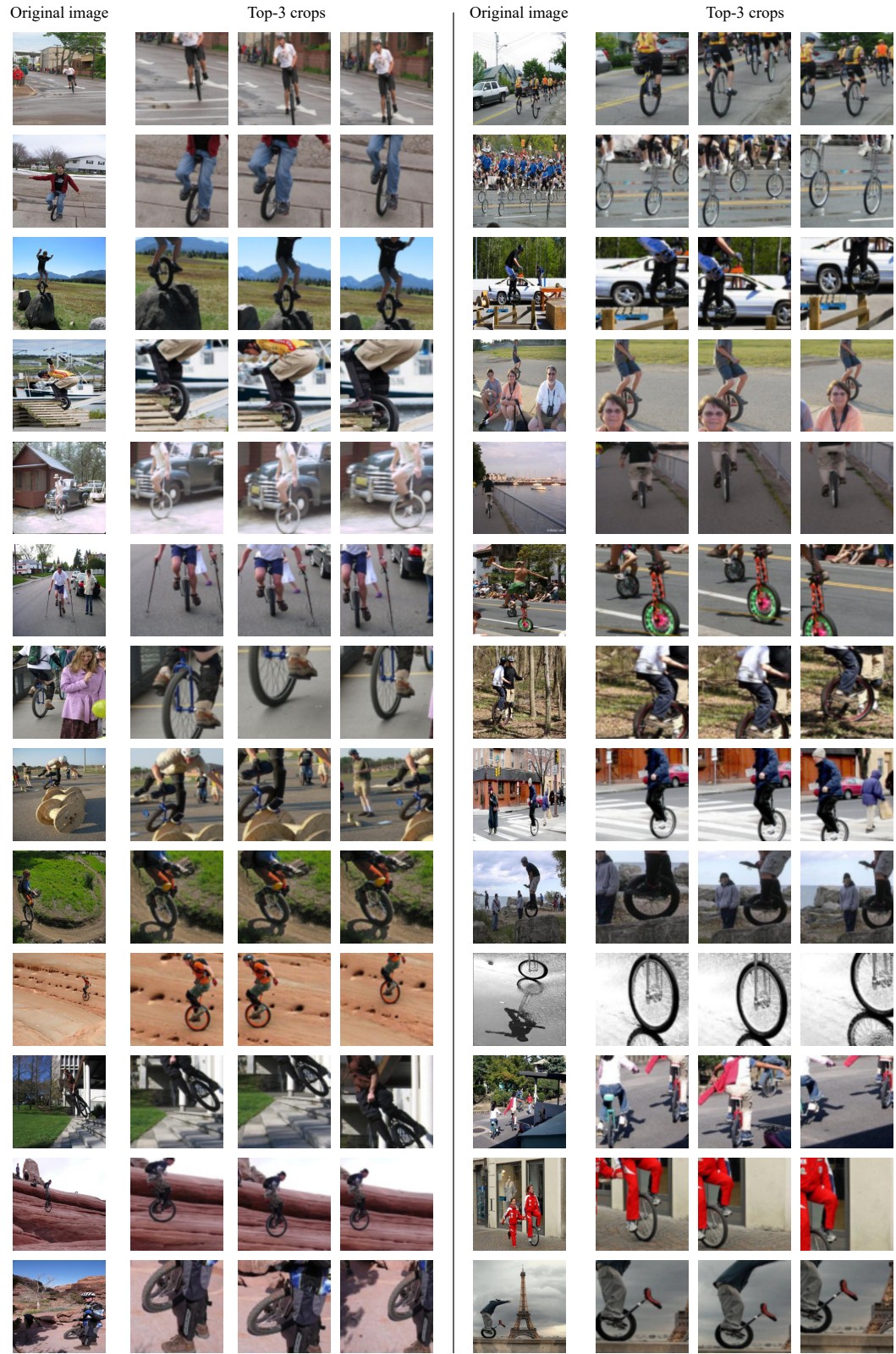

Figure 9: Visulization results of COS algorithm on class ***unicycle***.

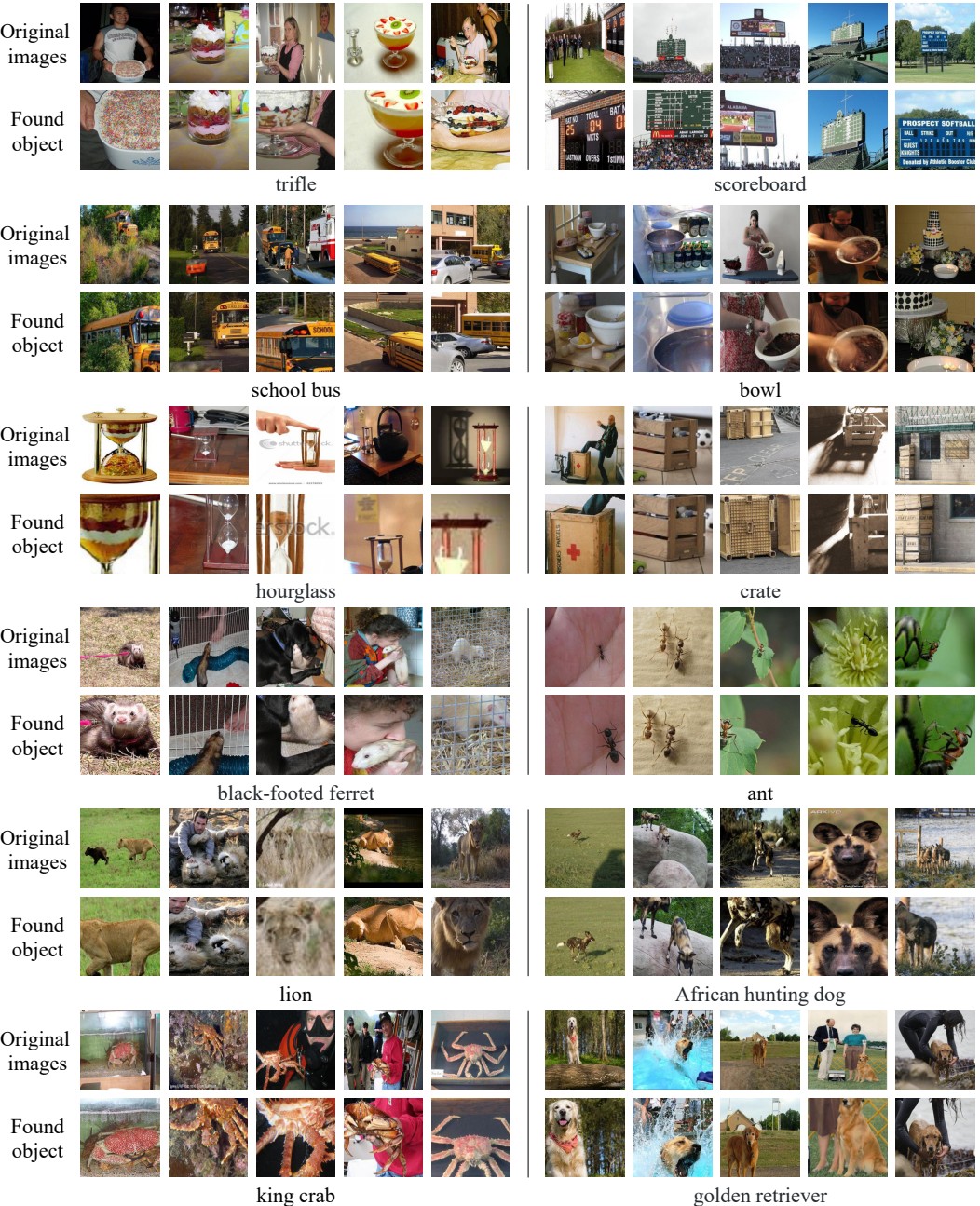

Figure 10: Additional visualization results of the first step of SOC algorithm. In each group of images, we show a 5-shot example from one class.