# OpenReview forum: "Rectifying the Shortcut Learning of Background for Few-Shot Learning"
_NeurIPS.cc/2021/Conference — NeurIPS 2021 Poster_

### Official Review · Reviewer_ua3N · 2021-07-12

**Rating:** 5
**Confidence:** 5

**Summary:**

The paper proposed a few-short learning (FSL) method by excluding the negative impact of background noise common in the existing FSL framework. A few separate stages are employed here to achieve the goal: 1) Clustering-based Object Seeker: a pretraining stage to pretrain the network and hence the feature space of the foreground objects; 2) Shared Object Concentrator (SOC): get the representation of the foreground of each class (which is then used in test time to determine the foreground of a test image). The method showed good empirical results on selected datasets.

**Limitations And Societal Impact:**

Limitations have been discussed in the work. No negative societal impact as far I can think of.

**Main Review:**

The originality of the work is moderate - Specific steps of training and extracting foreground features are integrated into the existing FSL framework. The paper is well-written in general except for a few issues mentioned below. The significance of the work though is work debating and a few major questions are listed here.

The key issue of the work resides in the test time of the method, i.e. using multiple crops to aggregate the scores of each class. I am sure this step can improve the final performance, however it is dangerously close to the common trick of test time augmentation to boost the model performance in practice. Simply put, for other few-shot methods, if we simply do multiple crops and averaging the scores as the final output, it is very likely that most of them, if not all, would see some performance improvement. If we further do a clever weighted sum, e.g. assign a weight based on the similarity of crops to the images of a given class in the training data, the performance may get further improved. In other words, the setup here is not persuasive enough to me that the improvement is mainly caused by rectifying the short learning of background - which is the main conclusion of the paper, although I’d be very much inclined to believe so in theory.

Secondly, the reason why the miniImageNet is not directly used in the comparison is worth a detailed explanation. Most of the SOTA methods report performance on miniImageNet directly, yet here a subset of miniImageNet is deliberately constructed for comparison. Why add this additional step into the experiment preparation?

Additionally, a large portion of the work tried to construct an effective foreground detector and tried to persuade the audience that the proposed method: random cropping + contrastive learning + clustering solved the problem well. It could easily trigger the curiosity of an audience that why not try some existing saliency detector and combine it with the few-shot framework? Does the proposed foreground detector work better than the existing saliency detectors? If so, is it because of better foreground detection results or because of the inherent pretraining step? A further discussion would be helpful for the audience to understand better the value of this work.


A couple of relatively minor issues that worth consideration:

1. In the Step 2 of 4.2, the meaning of \alpha^{j-1} and \beta^{n-1} is unclear. Specifically, the meaning of the superscript here could be further clarified especially if alpha and beta are all fixed in the paper?


2. Considering the numerous separate steps in the entire workflow, it would be helpful to show some failure cases and try to attribute the cause to the failure of individual steps.


**Time Spent Reviewing:**

1.5 hrs

---

> ### Author Response · Authors · 2021-08-09
> **Response to Reviewer ua3N**
>
> We thank the reviewer for the valuable detailed review. Our responses to the mentioned questions are below:
>
> ### **Q1: The key issue of the work resides in the test time of the method, i.e. using multiple crops to aggregate the scores of each class.**
>
> **A1**: We agree with the reviewer that this is one of the factors that might contribute to the improvement of the evaluation performance. For fair comparison and to better clarify the influence of our SOC algorithm, we have included additional experiments about the influence of multi-cropping. We implemented several Few-Shot Learning methods using multi-cropping during evaluation. Specifically, as the reviewer has suggested, for all methods except DeepEMD, we average the feature vectors of 7 crops and use the resulted averaged feature for classification. For DeepEMD, we notice that they also report performance using multi-cropping during the evaluation stage, thus we follow the method in the original paper. And it is noticed that its EMD classifier indeed shares a high similarity with your **clever weighted sum** suggestion. In this method, each crop is assigned a weight that is computed based on the similarity to crops from other images of the same class in the training data, and the weighted crops are used for solving an optimal transport problem to find correspondence between images. We report the results in the following Table. As a reference of upper bound, we have also included the performance of using the ground truth foreground. We denote multi-cropping as **MC**.
>
> |Method|1-shot(no MC->MC)|5-shot(no MC->MC)|
> |:-|:-|:-|
> |PN[2]|60.19->63.97|75.50->78.90|
> |Baseline[3]|60.93->63.83|78.46->81.38|
> |CC[4]|62.67->64.41|80.22->82.74|
> |Meta-baseline[5]|62.65->65.31|79.10->81.26|
> |RFS-distill[6]|63.00->65.02|79.63->82.04|
> |FEAT[7]|66.45->68.03|81.94->82.99|
> |DeepEMD[1]|66.61->67.63 | 82.02->83.47|
> |S2M2_R[8]|64.93->66.97|83.18->84.16|
> |COS|65.05->67.23|81.16->82.79|
> |COSOC|**69.28**(with MC)|**85.16**(with MC)|
> |COS+groundtruth|71.36->72.71|86.20->87.43|
>
> The results show that multi-cropping can improve FSL models by 1-3 points, and the improvement tends to be marginal when the baseline performance becomes higher. Moreover, the improvement is smaller in 5-shot settings. **Plugging this improvement into all methods listed in Table 2 of the main paper, we can see that our method still outperforms or be on par with SOTA results. On the other hand, our results are still significantly better than our COS baseline and are close to the upper bound which is obtained with the ground-truth foreground**.  This shows the effectiveness of our SOC algorithm. We will implement other FSL methods with multi-cropping at the evaluation stage and include the comparisons in our revised version of this paper.
>
>
> ### **Q2: The reason why the miniImageNet is not directly used in the comparison is worth a detailed explanation.**
>
> **A2**: We are sorry for not making this clear. The main results presented in Table 2 are obtained using the **original full miniImageNet** for fair comparison. We construct a subset but not all of miniImageNet mainly for ease of empirical investigation in Section 3---Manually cropping all images for constructing ground-truth foreground version of the full miniImageNet is time-consuming, so we choose parts of it to quickly obtain similar observations.
>
>
> ### **Q3: Why not try some existing saliency detector and combine it with the few-shot framework? Does the proposed foreground detector work better than the existing saliency detectors?If so, is it because of better foreground detection results or because of the inherent pretraining step?**
>
> **A3-1**: Nice suggestion! The observations in our paper pave a new way towards improving FSL by rectifying shortcut learning of background, which can be implemented using any effective unsupervised foreground detection methods. Our method is only one possible way. Given the upper bound performance in Table 1 in the main paper, we believe there is room in the future to improve final performance by using a more accurate foreground detector or a feature extractor with better main object discrimination ability. For comparisons with our method as the reviewer's suggestion, we implement three classical unsupervised saliency methods--**RBD[9], FT[10], MBD[11]**--to extract foreground objects from images. Specifically, for each image, a saliency map is obtained by the corresponding saliency method, followed by cropping out low-saliency regions. The cropping threshold is specially tuned.
>
> (1) Pretraining Stage. We use the fusion strategy with probability of 0.5 to train CC with crops obtained by unsupervised saliency methods. All models are trained **from scratch (including COS)** using the same training settings, which excludes the influence of Exampler-pretraining initialization. The results are shown below.
>
> |Method|1-shot|5-shot|
> |:-|:-|:-|
> |CC|62.67|80.22|
> |CC+RBD|63.24|80.45|
> |CC+MBD|61.50|79.12|
> |CC+FT|62.71|80.06|
> |CC+COS|64.76|81.18|
>
> (2) Evaluation Stage. We replace the original images with crops obtained by unsupervised saliency methods directly for classification. For fair comparison, we use the same feature extractor (COS) and also report performance using multi-cropping(MC) for all methods. The evaluation results are shown below.
>
> |Method|1-shot|5-shot|
> |:-|:-|:-|
> |COS(+MC)|65.05(67.23)|81.16(82.79)|
> |COS+RBD(+MC)|66.05(67.03)|81.88(82.57)|
> |COS+MBD(+MC)| 60.52(62.98)|77.12(79.56)|
> |COS+FT(+MC)| 62.87(64.74)| 78.79(80.74) |
> |COS+SOC|69.28|85.16|
> |COS with ground truth(+MC)|71.36(72.71)|86.20(87.43)|
>
> The results show that: (1) Our method performs consistently much better than considered unsupervised saliency detection methods. (2) The performance of different unsupervised saliency detection methods varies. While RBD gives a small improvement, MBD and FT have negative effects on the performance. The performance severely depends on the effectiveness of unsupervised saliency detection methods, and is very sensitive to the cropping thereshold. Intuitively speaking, saliency detection methods focus on noticeable objects in the image, and might fail when there is another irrelevant salient object in the image(e.g., a man is walking a dog. Dog is the label, but the man is of high salience). On the contrary, our method focuses on **shared** objects across images in the same class, thereby avoiding this problem. In addition, our COS algorithm has the ability to dynamically assign foreground scores to different patches, which reduces the risk of overconfidence. Given the upper bound with ground truth foreground, we believe there is room to improve and there can be other more effective approaches in the future!
>
> **A3-2**: To further exclude the concerns that the improvement of COS comes from the exampler-pretraining step, we show more ablation studies below.
>
> |Method|1-shot $D_v-Ori$|5-shot $D_v-Ori$|1-shot $D_v-FG$|5-shot $D_v-FG$|
> |:-|:-|:-|:-|:-|
> |CC|62.67|80.22|66.69|82.86|
> |Exampler|61.14|78.13|70.14|85.12|
> |CC+COS(from scratch)|64.76|81.18|71.13|86.21|
> |CC+COS(finetune)|65.05|81.16|71.36|86.20|
>
> From the above results, it can be seen that the effect of finetuning from Exampler is marginal. We adopt it mainly for a much faster converging speed, i.e., it takes 20 epochs for finetuning (used in our experiments) while 60 epochs from scratch.
>
> ### **Q4:In the Step 2 of 4.2, the meaning of $\alpha^{j-1}$ and $\beta^{n-1}$ is unclear.**
>
> **A4**: Sorry for this confusion.  Here $j-1$ and $n-1$ indicates powers, not superscripts. When $\alpha$ and $\beta$ are smaller than one, the values become smaller with the increase of $j$ and $n$. Therefore the values judge the importance or belief of different patches to be foreground objects. We will add more details in our revised version.
>
> ### **Q5:Considering the numerous separate steps in the entire workflow, it would be helpful to show some failure cases and try to attribute the cause to the failure of individual steps.**
>
> **A5**: Thanks for your suggestion! We agree that it is important to find and analyze failure cases in the algorithm, giving potential directions for further improvement. We will dive into the two steps of our model--COS and SOC, find potential failure cases, and include relevant discussions in the main paper or appendix. We have done preliminary experiments, and found one issue that the best cropping size differs across classes, making it not easy to determine a fixed cropping size used for all data. Some objects from one class are often small in the image (e.g., birds), while some objects from another class take up a larger ratio of space (e.g., buildings). If the cropping size is not appropriate, it is possible that background information mixes in, or only parts of the object are covered. In addition, a too small cropping size increases the probability that no crops interleave with the main object. It is future work to solve this problem by first crudely locating potential objects in the images with detection/segmentation algorithms--to automatically decide the cropping size--and then using these detected crops for further processing.
>
> [1] Zhang et al., DeepEMD: Differentiable Earth Mover's Distance for Few-Shot Learning.
>
> [2] Snell et al., Prototypical Networks for Few-shot Learning.
>
> [3] Chen et al., A Closer Look at Few-shot Classification.
>
> [4] Gidaris et al., Dynamic few-shot visual learning without forgetting.
>
> [5] Chen et al., A New Meta-Baseline for Few-Shot Learning.
>
> [6] Tian et al., Rethinking few-shot image classification: A good embedding is all you need?
>
> [7] Ye et al., Few-shot learning via embedding adaptation with set-to-set functions.
>
> [8] Mangla et al., Charting the Right Manifold: Manifold Mixup for Few-shot Learning.
>
> [9] Zhu et al., Saliency Optimization from Robust Background Detection.
>
> [10] Achanta et al., Frequency-tuned Salient Region Detection.
>
> [11] Zhang et al., Minimum Barrier Salient Object Detection at 80 FPS.

---

> > ### Comment · Reviewer_ua3N · 2021-08-25
> > **Thank you for the detailed response!**
> >
> > It has been very helpful and constructive.

---

> > > ### Author Response · Authors · 2021-08-25
> > > **Thank you for acknowledging our response**
> > >
> > > We are glad that our response has addressed the concerns of the reviewer. Indeed, the suggestions of the alternative idea of using saliency detection methods by the reviewer are very motivating, which, in fact, have not been proposed in the literature of Few-Shot Learning before. Although some of the methods we've tried do not work better than our proposed method, they do form a comparable baseline and benefit our understanding of the role of background knowledge in Few-Shot Learning. We will acknowledge these points in the paper.
> > >
> > > If you have any further questions or suggestions, we are very happy to discuss with you.

---

> > > > ### Comment · Reviewer_ua3N · 2021-09-01
> > > > **Thank you for the detailed response!**
> > > >
> > > > Based on the discussion and reviews, I'd like to increase my rating to "Marginally below the acceptance threshold". I feel the paper as its current content needs some edition before publish to give the full picture to the audience, especially about the discussions on test time augmentation in the rebuttal. I do appreciate the author's constructive response.

---

> > > > > ### Author Response · Authors · 2021-09-02
> > > > > **Thanks for additional comments!**
> > > > >
> > > > > We agree with reviewer that our paper needs to add some additional results (The impact of multi-cropping) based on the rebuttal, and we promise that we will carefully revise the paper once the revision is admitted. Specifically, we will replace the original results of other methods in Table 2 of the paper with those obtained with multi-cropping for fair comparison, with proper interpretations. Additionally, in order to give a full picture of our method to the audience, we will add the comparison of our method with unsupervised saliency methods and additional ablation studies (The impact of pre-training) in the appendix.

---

### Official Review · Reviewer_z1Ft · 2021-07-13

**Rating:** 7
**Confidence:** 4

**Summary:**

This paper dives into the importance of learned “shortcuts” based on background similarly across the training samples in few-shot learning scenarios, at the detriment of generalization to images with a “category gap” - a background that does not match what was seen in training. They explore the effect of background on recognition and propose two simple algorithms based on contrastively-learned features (one method at train time, one at test) to find foreground vs background and learn more generalizable representations with little data. At training time they use clustering over contrastive-pretrained features extracted from a set of crops from images across the training data for each class to find crops that are highly likely to contain foreground and use those crops instead of the image (with some probability, based on the “foreground score” of the crop). At test time a representative feature per-class is generated using the training data, and simple feature matching is used to determine the class.

**Ethical Concerns:**

I do not see ethical concerns with this paper

**Limitations And Societal Impact:**

Beyond the known and problematic biases in the underlying datasets they use, I do not see potential for negative societal impact from this work

**Main Review:**

This paper was an enjoyable read - It was well laid out and clearly written. The problem formulation was justified with the analysis in section 3.1, the methods were simple, intuitive, and explained clearly and the results outperformed the methods they compared against. I like that the methods were so simple (feature matching!) and relied heavily on available and powerful contrastively-learned feature representations. There is possibly further room to improve by extending these simple approaches in the future.

Here’s an earlier paper that seems quite relevant, it points out performance drops when moving to novel image backgrounds: https://arxiv.org/abs/1807.04975

This method has a strong inter-class visual similarity assumption, are there categories which might break this? Did you see any consistent failure modes of the method? Possibly worth mentioning in discussion.

In line 109: “crop each image manually” - how exactly? Do you take the largest bounding box inside the object or make a per-pixel segmentation mask?


**Time Spent Reviewing:**

2

---

> ### Author Response · Authors · 2021-08-09
> **Response to Reviewer z1Ft**
>
> We thank the reviewer for the positive review and insightful suggestions. Answers to specific points are below:
>
> ### **Q1:Here’s an earlier paper that seems quite relevant, it points out performance drops when moving to novel image backgrounds.**
>
> **A1**: Thanks for mentioning this paper! The authors in this paper utilized camera traps to investigate how performance drops when adapting classifiers to unseen cameras with novel backgrounds. It is an excellent experimental setup that explores the effect of the class-independent background on classification performance, *i.e.*, background is changing from training to testing while categories remain unchanged. Although the problem is also about image background, there is no shortcut learning of background under this setting. This is because under each training camera trap, the classifier must distinguish different categories from the same background. Therefore the background knowledge is not useful for predictions of training images. Instead, the learning signal  during training pushes the classifier towards ignoring each specific background. The difficulties under this setting lie in the domain shift challenge---the feature extractor that is familiar with how to ignore existing backgrounds does not know how to deal with the unseen backgrounds. We will include the above discussions in our revised version.
>
>
>
> ### **Q2: This method has a strong intra-class visual similarity assumption, are there categories which might break this? Did you see any consistent failure modes of the method?**
>
> **A2-1**: In general, the definition of "category" ensures the intra-class visual similarity among objects from the same class, thus we expect there is nearly no categories to break this. We have empirically checked the dataset and found it true in most cases. However, there might be possible "outlier samples" in each class to break this. After having a look at images in miniImageNet, we find some examples that contain foreground distinctly visually different from others, e.g. n0296619300000058.jpg, n0274717700000027.jpg (We will display some of them in the appendix and include the corresponding discussions). For these images, our COS algorithm will assign low foreground confidence scores to all of their crops, which means they will be sample with low sampling probability. Therefore, our approach automatically rectifies the wrong belief.
>
> **A2-2**: Aside from the aforementioned low intra-class visual similarity problem, we have done preliminary experiments and found one issue that the best cropping size differs across classes, making it not easy to decide on a fixed cropping size used for all data. Some objects from one class are often small in the image (e.g., birds), while some other objects take up a larger ratio of space (e.g., buildings). If the cropping size is not appropriate, it is possible that background information mixes in, or only parts of the object are covered. It is our future work to solve this problem by first crudely locating potential objects in the images by detection/segmentation algorithms--to automatically decide the cropping size--and then using these detected crops for further processing.
>
> ### **Q3:In line 109: “crop each image manually” - how exactly? Do you take the largest bounding box inside the object or make a per-pixel segmentation mask?**
>
> **A3**: We take the largest bounding box inside the object.

---

> > ### Comment · Reviewer_z1Ft · 2021-08-16
> > **Thank you for your response!**
> >
> > I appreciate the clarifications.

---

> > > ### Author Response · Authors · 2021-08-25
> > > **Thank you for your comment.**
> > >
> > > We really appreciate your constructive review and precious time.

---

### Official Review · Reviewer_2di6 · 2021-07-15

**Rating:** 7
**Confidence:** 4

**Summary:**

In this work, the authors conducted an emperical analysis and revealed that image background serves as a source of shortcut knowledge which is harmful for few-shot learning task. To alleviate this, they proposed a two-stage method, COSOC, that employs contrastive learning to draw the model's attention to image foreground during both pre-training and evaluation stage. Given the assumption that foreground objects from different images of the same class share more similar patterns as background, in the pre-training stage, a clustering is performed on randomly cropped patches to identify crops that best represent the foreground objects in the image. Then, fusion sampling is adopted, i.e. these crops are used to replace original image with a certain probability based on its foreground score during pre-training. In the evaluation stage, mean features maximally representing the shared contents among training images from the same class is obtained. Given this, the foregrounds in testing images can be located via matching.

**Limitations And Societal Impact:**

Please refer to "Main Review" section above

**Main Review:**

Strengths
- The overall writing is clear and easy to follow.
- Although the idea of removing background to avoid learning shortcut knowledge is intuitive, its realization is however non-trivial. The authors provide a feasible solution to this, where its effectiveness is well proven by extensive experiments.

Questions
- The fusion strategy selects either the original image or randomly cropped patch with a certain probability. Why not using the top K crops discovered in step 4 in L202 to replace the randomly cropped patch? This also better simulates the training setting in Sec 3.1.
- In the evaluation stage, instead of computing features $\omega$ that minimizes the sum of pair-wise distances between crops from different images within the same class, why not performing clustering on the training images to extract reliable crops representing the foregrounds similar to what's being done in the pre-training stage for matching?
- A simple yet possibly strong baseline could be running unsupervised saliency detection to extract the most salient region in an image, followed by cropping to obtain patches without background. This also eliminates the need of having large number of crops to avoid missing the foreground. The authors could consider comparing with this.

**Time Spent Reviewing:**

6

---

> ### Author Response · Authors · 2021-08-09
> **Response to Reviewer 2di6**
>
> We thank the reviewer for the positive review and insightful questions. Answers to specific points are below:
>
> ### **Q1:Why not using the top K crops discovered in step 4 in L202 to replace the randomly cropped patch?**
>
> **A1**: Sorry for this typo. Note that the subscript $i$ in line 212 is randomly sampled from 1~K. Therefore we implement exactly what you describe (use top K crops as foreground candidates). We will fix the typo by changing $p_{n,m}$ to $p_{n, \beta_i}$ in line 211.
>
>
> ### **Q2:During evaluation, why not performing clustering on the training images to extract reliable crops representing the foregrounds similar to what's being done in the pre-training stage for matching?**
>
> **A2**: The clustering method has a severe problem when the number of images becomes small--it is more likely that different crops come from the same image.  Crops from the same image have much higher probabilities to share high similarities, making it possible that each cluster only has crops from one same image. However, what we need are similar crops from different images, thus simple clustering methods are inappropriate to directly apply in few-shot evaluation stage. On the contrary, our SOC algorithm explicitly seeks similar patches across images.
>
>
> ### **Q3: A simple yet possibly strong baseline could be running unsupervised saliency detection to extract the most salient region in an image, followed by cropping to obtain patches without background.The authors could consider comparing with this.**
>
> **A3**: Thanks for this valuable suggestion! For comparisons with our method as the reviewer's suggestion, we have implemented three classical unsupervised saliency methods--**RBD[9], FT[10], MBD[11]**--to extract foreground objects from images. Specifically, for each image, a saliency map is obtained by the corresponding saliency method, followed by cropping out low-saliency regions. The cropping threshold is specially tuned.
>
> (1) Pretraining Stage. We use the fusion strategy with probability of 0.5 to train CC with crops obtained by unsupervised saliency methods. All models are trained **from scratch (including COS)** using the same training settings, excluding the influence of Exampler-pretraining. The results are shown below.
>
> |Method|1-shot|5-shot|
> |:-|:-|:-|
> |CC|62.67|80.22|
> |CC+RBD|63.24|80.45|
> |CC+MBD|61.50|79.12|
> |CC+FT|62.71|80.06|
> |CC+COS|64.76|81.18|
>
> (2) Evaluation Stage. We replace the original images with crops obtained by unsupervised saliency methods directly for classification. For fair comparison, we use the same feature extractor (COS) and also report performance using multi-cropping(MC) for all methods. The evaluation results are shown below.
>
> |Method|1-shot|5-shot|
> |:-|:-|:-|
> |COS(+MC)|65.05(67.23)|81.16(82.79)|
> |COS+RBD(+MC)|66.05(67.03)|81.88(82.57)|
> |COS+MBD(+MC)| 60.52(62.98)|77.12(79.56)|
> |COS+FT(+MC)| 62.87(64.74)| 78.79(80.74) |
> |COS+SOC|69.28|85.16|
> |COS with ground truth(+MC)|71.36(72.71)|86.20(87.43)|
>
> The results show that: (1) Our method performs consistently much better than the listed unsupervised saliency methods. (2) The performance of different unsupervised saliency methods varies. While RBD gives a small improvement, MBD and FT have negative effect on the performance. The performance severely depends on the effectiveness of unsupervised saliency methods, and is very sensitive to the cropping threshold.
>
> Intuitively speaking, saliency detection methods focus on noticeable objects in the image, and might fail when there is another irrelevant salient object in the image(e.g., a man is walking a dog. Dog is the label, but the man is of high salience). On the contrary, our method focuses on **shared** objects across images in the same class, thereby avoiding this problem. In addition, our COS algorithm has the ability to dynamically assign foreground scores to different patches, which reduces the risk of overconfidence. One of our main contributions is paving a new way towards improving FSL by rectifying shortcut learning of background, which can be implemented using any effective methods. Given the upper bound with ground truth foreground, we believe there is room to improve and there can be other more effective approaches in the future!
>
> [1] Zhu et al., Saliency Optimization from Robust Background Detection. CVPR 2014.
>
> [2] Achanta et al., Frequency-tuned Salient Region Detection. CVPR 2009.
>
> [3] Zhang et al., Minimum Barrier Salient Object Detection at 80 FPS. ICCV 2015.

---

> > ### Comment · Reviewer_2di6 · 2021-08-24
> > **Response to Authors Rebuttal**
> >
> > Thank you for your thorough response to my review, and including more experiments with unsupervised saliency detection. I appreciate the additional information provided.

---

> > > ### Author Response · Authors · 2021-08-25
> > > **Thank you for the comment.**
> > >
> > > We really appreciate your constructive review and your precious time.

---

### Official Review · Reviewer_Q4tA · 2021-07-21

**Rating:** 5
**Confidence:** 5

**Summary:**

This paper presents a new few-shot learning framework COSOC that can focus more on foreground objects at both pre-training and evaluation stage. COSOC is a two-stage algorithm. At the pretraining stage, a clustering-based object seeker (COS) module is used to force the pretraining model to focus on found foreground objects by a fusion sampling strategy. At the evaluation stage, a shared object concentrator (SOC) module is used to seek for shared contents and filter out background, and then the foreground of testing images is matched with the recognized foreground objects of each class. Experiments on two benchmarks are conducted to verify the effectiveness of the presented method.

**Ethical Concerns:**

There are no obvious ethical concerns on this work.

**Limitations And Societal Impact:**

Yes

**Main Review:**

The strengths of this work are as follows.

+This paper considers that background information is easy to be used as a shortcut to distinguish images and constructs a small dataset D_new with proper experiments to analyze the adverse effect of background information in few-shot learning.

+This paper presents a new few-shot learning framework COSOC.

+The results seem good.

However, several major concerns should be addressed.

-More details should be provided for the training of contrastive learning models (Exampler used in this paper, a modified version of MoCo). It raises a major concern on the baseline. How the model is initialized? How many training epochs are used? The training is commonly expensive, and how much time is used? Does the training follow the original paper?

-As shown by the results, the presented method achieves more performance gain in the 1-shot setting. But it is really in doubt that the presented method can locate the foreground by using V randomly crops (no clustering is used in 1-shot setting, as stated in the paper). This is a major problem.


**Time Spent Reviewing:**

30

---

> ### Author Response · Authors · 2021-08-09
> **Response to Reviewer Q4tA**
>
> We thank the reviewer for providing insightful comments. Our responses to the mentioned questions are below:
>
> ### **Q1:More details should be provided for the training of contrastive learning models (Exampler used in this paper, a modified version of MoCo). It raises a major concern on the baseline. How the model is initialized? How many training epochs are used? The training is commonly expensive, and how much time is used? Does the training follow the original paper?**
>
>
>
> #### **A1-1: Concerns about details of pretraining Exampler and time complexity.**
>
> We follow the training details in the original paper of Exampler except that we train on miniImageNet instead of ImageNet, and use ResNet12 as the backbone instead of ResNet50.  Specifically, we use a queue with the length of 32768. The momentum is set to 0.999 and the temperature is 0.07. We set the output dimension of MLP as 128. We run it for 1000 epochs on miniImageNet using two Nvidia 1080Ti within approximately 2 days.
>
> To make it clearer, we give additional comparisons on training time complexity with some other FSL models based on the estimated time required for training (including the pre-training time). All of the experiments are based on the given official public codes. (Reasons for a long-time running: (1) DeepEMD: time-comsuming gradient backpropagation of EMD distance and an inherent pretraining step before meta-learning. (2)S2M2: a deeper WRN network, slow convergence (total 600 epochs) and extra 4 copies of each sample sent to the network (for rotation task). (3) IER-distill: extra 32 copies of samples sent to the network (for various tasks and a student network) and an extra distillation round.)
>
> |Method|Estimated training hours using two GPUs|
> |:----|:----|
> |DeepEMD[1]|200|
> |S2M2_R[2]|205|
> |IER-Distill[3]|130|
> |COSOC|50|
>
> #### **A1-2: Concerns about the influence of Exampler-pretrained initialization.**
>
> Our model is initialized from the Exampler-pretrained backbone. It is very common for Few-Shot Learning models to be initialized from a pretrained backbone \[1][4]\[5][6] for performance improvement. However, here we are not intended to use the pretrained Exampler to boost performance, but just to accelerate convergence of training---it takes 20 epochs for finetuning (used in our experiments) while 60 epochs for training from scratch. In the following additional ablation study, we can see that the effect of finetuning from Exampler is marginal.
>
> |Method|1-shot $D_v-Ori$|5-shot $D_v-Ori$|1-shot $D_v-FG$|5-shot $D_v-FG$|
> |:----|:----|:----|:----|:----|
> |CC|62.67|80.22|66.69|82.86|
> |Exampler|61.14|78.13|70.14|85.12|
> |CC+COS(from scratch)|64.76|81.18|71.13|86.21|
> |CC+COS(finetune)|65.05|81.16|71.36|86.20|
>
>
> ### **Q2:It is really in doubt that SOC can locate the foreground by using V randomly crops in one-shot setting.**
>
> **A2**: At first, we want to mention that there is more room for improvement on 1-shot accuracy than that on 5-shot accuracy in Few-Shot Learning, since 1-shot accuracy is typically much lower. **A better metric that measures improvement may be the percentage of error that is reduced by the model**. Under this metric, the improvement of using SOC for evaluation is **12.13%** and **20.06%** for 1-shot and 5-shot settings respectively (from Table 1), which we hope dispels your doubts.
>
> Although there is no search for shared information among training images in 1-shot setting,  the matching between the crops in training and testing images helps to locate the foreground objects in both images as well. The top-matched crops share high similarities. They are more likely to be foreground objects and have higher weights for comparison. In fact, our method **in the one-shot setting** is in spirit similar to several recent SOTA FSL methods[1]\[7][8], in which the models seek correspondences between pairs of images by computing feature similarity. In both our and their methods, the background can be in principle implicitly removed due to the dissimilarity between different background of images. However, their methods use spatial features obtained from a ResNet for seeking spatial correspondence,  where the output features share a large amount of mixed spatial information due to a relatively large receptive field (Particularly, the receptive field of each feature of ResNet-12 is the whole image). This makes it impossible to completely remove background, whereas our model explicitly removes part of the original image by random cropping, avoiding such problems. On the other hand, we check a bunch of random crops of images, and most of these images have at least one random crop that properly bound all or parts of the foreground object, ensuring the "true" correspondence of foreground exists.
>
> [1] Zhang et al., DeepEMD: Differentiable Earth Mover's Distance for Few-Shot Learning. CVPR 2020.
>
> [2] Mangla et al., Charting the Right Manifold: Manifold Mixup for Few-shot Learning. WACV 2020.
>
> [3] Rizve et al., Exploring complementary strengths of invariant and equivariant representations for few-shot learning. CVPR 2021.
>
> [4] Zhang et al., IEPT: instance-level and episode-level pretext tasks for few-shot learning. ICLR 2021.
>
> [5] Fei et al., MELR: meta-learning via modeling episode-level relationships for few-shot learning. ICLR 2021.
>
> [6] Xu et al., Attentional Constellation Nets for Few-Shot Learning. ICLR 2021.
>
> [7] Xu et al, Learning dynamic alignment via meta-filter for few-shot learning. CVPR 2021.
>
> [8] Doersch et al., Crosstransformers: spatially-aware few-shot transfer. NIPS 2020.

---

> > ### Comment · Reviewer_Q4tA · 2021-09-02
> > **Follow-up**
> >
> > Thanks for the response. In experiments, is Exampler only trained on the whole miniImageNet? But two datasets are evaluated in the paper, do you directly apply the trained Exampler to tieredImageNet?

---

> > > ### Author Response · Authors · 2021-09-02
> > > **Thanks for your reply!**
> > >
> > > We are sorry for the confusion. The Exemplar is trained on tieredImageNet and miniImageNet separately.
> > >
> > > If you have any additional concerns please let us know and we would be happy to follow up. Thanks!

---

> > ### Comment · Reviewer_Q4tA · 2021-09-03
> > **Thanks for the response**
> >
> > After reading the other reviewers’ comments and the rebuttal, I will keep my initial rating. The rebuttal has addressed part of my concerns, but the main concerns still exist.
> > 1. The handling of 5-shot setting fits the proposed idea more, while the 1-shot setting follows somewhat different procedure from the main idea. However, the results seem that it achieves more performance gain in the 1-shot setting. The two are suggested to be better unified.
> > 2. I still have doubt in the training of Exampler. I think its performance is likely to affect the results a lot, but these details cannot be well clarified in the paper and in the rebuttal at the first time, which makes some statements less convincing. Moreover, I also consider that training Exampler on the whole miniImageNet or tiredImageNet is not appropriate, since it makes the Exampler ‘’see’’ the data of novel examples in representation learning and can improve the results of clustering and other components in the method.

---

> > > ### Author Response · Authors · 2021-09-03
> > > **Thanks for your comment!**
> > >
> > > 1. The reason why it seems our method under 1-shot setting improves more has been explained in the rebuttal. **Here we want to emphasize that, due to the essential difference between 1-shot and multi-shot problem (whether need to aggregate class information for metric-based comparison), the inconsistency between above two settings generally exists in various metric-based methods for few-shot learning**. For example, In DeepEMD [1], they design a Structured Fully Connected Layer, which is customized for multi-shot setting; the main idea of Prototypical Network [2] is to average the training features of the same class into a prototype, which also makes no sense in 1-shot setting. The same exists in a large body of following works that also utilize the average operation for class agrregation [3-10]. The average operation does not consider the importance of different components in images, and we make a step forward to reveal the harmfulness of background and try to solve it. Here, to aggregate shared information among training images, an "operation" customized only for multi-shot problem is indispensable, resulting in the inconsistency under 1-shot and multi-shot settings. While step 1 of SOC under 1-shot setting makes no sense, step 2 of SOC shares the same motivation from different aspects (seeking shared information between training and testing images), thus the method under 1-shot setting does not deviate from the main idea.
> > >
> > > 2. We clarify that the word "whole" means that the Exemplar is only trained on the whole **training set** of miniImageNet or tieredImageNet instead of $D_B-new~$ subset in our paper, thereby never seeing novel examples in representation learning. As for the effect of pretraining of Exemplar on the performance, we have shown in the rebuttal that its help is incremental. This is natural because the performance of Cosine Classifier is originally much better than Exemplar. The use of Exemplar is mainly for filtering out background of training images. We will clarify more in the paper about this potential confusion.
> > >
> > > [1] Zhang et al. DeepEMD: Differentiable Earth Mover's Distance for Few-Shot Learning. CVPR 2020.
> > >
> > > [2] Snell et al. Prototypical Networks for Few-shot Learning. NIPS 2017.
> > >
> > > [3] Ye et al. Few-shot learning via embedding adaptation with set-to-set functions. CVPR 2020.
> > >
> > > [4] Zhang et al. IEPT: instance-level and episode-level pretext tasks for few-shot learning. ICLR 2021.
> > >
> > > [5] Doersch et al. Crosstransformers: spatially-aware few-shot transfer. NIPS 2020.
> > >
> > > [6] Xu et al. Learning dynamic alignment via meta-filter for few-shot learning. CVPR 2021.
> > >
> > > [7] Wertheimer et al. Few-shot classification with feature map reconstruction networks. CVPR 2021.
> > >
> > > [8] Xu et al. Attentional constellation nets for few-shot learning. ICLR 2021.
> > >
> > > [9] Hou et al. Cross attention network for few-shot classification. NIPS 2019.
> > >
> > > [10] Oreshkin et al. TADAM: Task dependent adaptive metric for improved few-shot learning. NIPS 2018.

---

> ### Author Response · Authors · 2021-09-01
> **Response to Reviewer Q4tA (2)**
>
> Dear reviewer, we have tried to address your concerns in our earlier response. If you have any further questions or suggestions, we are very happy to discuss with you.

---

### Decision · Program_Chairs · 2021-09-27

**Decision:**

Accept (Poster)

**Comment:**

The authors provide a simple but effective technique for reducing bias towards background pixels, in a few-shot setting. They provided extensive data and explanations in their rebuttal. Though two of the reviewers remained unconvinced, the other two reviewers were very supportive. However, I believe that the simplicity of the method and it's clear effectiveness, along with the extensive empirical studies, makes it a good candidate to be accepted to Neurips 2021. The authors should (as they have themselves promised) revise the paper to incorporate the good comments from the reviewers.